# AC/DC: Alternating Compressed/DeCompressed Training of Deep Neural Networks

**Alexandra Peste** *
IST Austria

**Eugenia Iofinova**
IST Austria

**Adrian Vladu**
CNRS & IRIF

**Dan Alistarh**
IST Austria & Neural Magic

## Abstract

The increasing computational requirements of deep neural networks (DNNs) have led to significant interest in obtaining DNN models that are *sparse*, yet *accurate*. Recent work has investigated the even harder case of *sparse training*, where the DNN weights are, for as much as possible, already sparse to reduce computational costs during training. Existing sparse training methods are often empirical and can have lower accuracy relative to the dense baseline. In this paper, we present a general approach called Alternating Compressed/DeCompressed (AC/DC) training of DNNs, demonstrate convergence for a variant of the algorithm, and show that AC/DC outperforms existing sparse training methods in accuracy at similar computational budgets; at high sparsity levels, AC/DC even outperforms existing methods that rely on accurate pre-trained dense models. An important property of AC/DC is that it allows *co-training* of dense and sparse models, yielding accurate *sparse–dense model pairs* at the end of the training process. This is useful in practice, where compressed variants may be desirable for deployment in resource-constrained settings without re-doing the entire training flow, and also provides us with insights into the accuracy gap between dense and compressed models. The code is available at: `https://github.com/IST-DASLab/ACDC`.

## 1 Introduction

The tremendous progress made by deep neural networks in solving diverse tasks has driven significant research and industry interest in deploying efficient versions of these models. To this end, entire families of model compression methods have been developed, such as pruning [29] and quantization [22], which are now accompanied by hardware and software support [55, 8, 11, 43, 23].

Neural network pruning, which is the focus of this paper, is the compression method with arguably the longest history [38]. The basic goal of pruning is to obtain neural networks for which many connections are removed by being set to zero, while maintaining the network's accuracy. A myriad pruning methods have been proposed—please see [29] for an in-depth survey—and it is currently understood that many popular networks can be compressed by more than an order of magnitude, in terms of their number of connections, without significant accuracy loss.

Many accurate pruning methods require a fully-accurate, dense variant of the model, from which weights are subsequently removed. A shortcoming of this approach is the fact that the memory and computational savings due to compression are only available for the *inference*, post-training phase, and not during training itself. This distinction becomes important especially for large-scale modern models, which can have millions or even billions of parameters, and for which fully-dense training can have high computational and even non-trivial environmental costs [53].

One approach to address this issue is *sparse training*, which essentially aims to remove connections from the neural network as early as possible during training, while still matching, or at least approximating, the accuracy of the fully-dense model. For example, the RigL technique [16] randomly

---

*Correspondence to: Alexandra Peste <alexandra.peste@ist.ac.at>

35th Conference on Neural Information Processing Systems (NeurIPS 2021).

removes a large fraction of connections early in training, and then proceeds to optimize over the sparse support, providing savings due to sparse back-propagation. Periodically, the method re-introduces some of the weights during the training process, based on a combination of heuristics, which requires taking full gradients. These works, as well as many recent sparse training approaches [4, 44, 32], which we cover in detail in the next section, have shown empirically that non-trivial computational savings, usually measured in theoretical FLOPs, can be obtained using sparse training, and that the optimization process can be fairly robust to sparsification of the support.

At the same time, this line of work still leaves intriguing open questions. The first is *theoretical*: to our knowledge, none of the methods optimizing over sparse support, and hence providing training speed-up, have been shown to have convergence guarantees. The second is *practical*, and concerns a deeper understanding of the relationship between the densely-trained model, and the sparsely-trained one. Specifically, (1) most existing sparse training methods still leave a non-negligible accuracy gap, relative to dense training, or even post-training sparsification; and (2) most existing work on sparsity requires significant changes to the training flow, and focuses on maximizing global accuracy metrics; thus, we lack understanding when it comes to *co-training* sparse and dense models, as well as with respect to correlations between sparse and dense models *at the level of individual predictions*.

**Contributions.** In this paper, we take a step towards addressing these questions. We investigate a general hybrid approach for sparse training of neural networks, which we call *Alternating Compressed / DeCompressed (AC/DC)* training. AC/DC performs *co-training of sparse and dense models*, and can return both an accurate *sparse* model, and a *dense* model, which can recover the dense baseline accuracy via fine-tuning. We show that a variant of AC/DC ensures convergence for general non-convex but smooth objectives, under analytic assumptions. Extensive experimental results show that it provides state-of-the-art accuracy among sparse training techniques at comparable training budgets, and can even outperform *post-training* sparsification approaches when applied at high sparsities.

AC/DC builds on the classic *iterative hard thresholding (IHT)* family of methods for sparse recovery [6]. As the name suggests, AC/DC works by alternating the standard *dense* training phases with *sparse* phases where optimization is performed exclusively over a fixed *sparse support*, and a subset of the weights and their gradients are fixed at zero, leading to computational savings. (This is in contrast to *error feedback* algorithms, e.g. [9, 40] which require computing fully-dense gradients, even though the weights themselves may be sparse.) The process uses the same hyper-parameters, including the number of epochs, as regular training, and the frequency and length of the phases can be safely set to standard values, e.g. 5–10 epochs. We ensure that training ends on a *sparse* phase, and return the resulting *sparse* model, as well as the last *dense* model obtained at the end of a *dense* phase. This dense model may be additionally fine-tuned for a short period, leading to a more accurate *dense-finetuned* model, which we usually find to match the accuracy of the *dense baseline*.

We emphasize that algorithms alternating sparse and dense training phases for deep neural networks have been previously investigated [33, 25], but with the different goal on using sparsity as a regularizer to obtain *more accurate dense models*. Relative to these works, our goals are two-fold: we aim to produce highly-accurate, highly-sparse models, but also to maximize the fraction of training time for which optimization is performed over a sparse support, leading to computational savings. Further, we are the first to provide convergence guarantees for variants of this approach.

We perform an extensive empirical investigation, showing that AC/DC provides consistently good results on a wide range of models and tasks (ResNet [28] and MobileNets [30] on the ImageNet [49] / CIFAR [36] datasets, and Transformers [56, 10] on WikiText [42]), under standard values of the training hyper-parameters. Specifically, when executed on the same number of training epochs, our method outperforms all previous *sparse training* methods in terms of the accuracy of the resulting sparse model, often by significant margins. This comes at the cost of slightly higher theoretical computational cost relative to prior sparse training methods, although AC/DC usually reduces training FLOPs to 45–65% of the dense baseline. AC/DC is also close to the accuracy of state-of-the-art post-training pruning methods [37, 52] at medium sparsities (80% and 90%); surprisingly, it *outperforms* them in terms of accuracy, at higher sparsities. In addition, AC/DC is flexible with respect to the structure of the "sparse projection" applied at each compressed step: we illustrate this by obtaining *semi-structured* pruned models using the 2:4 sparsity pattern efficiently supported by new NVIDIA hardware [43]. Further, we show that the resulting sparse models can provide significant real-world speedups for DNN inference on CPUs [12].

An interesting feature of AC/DC is that it allows for accurate dense/sparse co-training of models. Specifically, at medium sparsity levels (80% and 90%), the method allows the co-trained dense

model to recover the dense baseline accuracy via a short fine-tuning period. In addition, dense/sparse co-training provides us with a lens into the training dynamics, in particular relative to the sample-level accuracy of the two models, but also in terms of the dynamics of the sparsity masks. Specifically, we observe that co-trained sparse/dense pairs have higher sample-level agreement than sparse/dense pairs obtained via post-training pruning, and that weight masks still change later in training.

Additionally, we probe the accuracy differences between sparse and dense models, by examining their "memorization" capacity [60]. For this, we perform dense/sparse co-training in a setting where a small number of valid training samples have *corrupted labels*, and examine how these samples are classified during dense and sparse phases, respectively. We observe that the sparse model is less able to "memorize" the corrupted labels, and instead often classifies the corrupted samples to their *true* (correct) class. By contrast, during dense phases model can easily "memorize" the corrupted labels. (Please see Figure 2b for an illustration.) This suggests that one reason for the higher accuracy of dense models is their ability to "memorize" hard-to-classify samples.

## 2 Related Work

There has recently been tremendous research interest into pruning techniques for DNNs; we direct the reader to the recent surveys of [21] and [29] for a more comprehensive overview. Roughly, most DNN pruning methods can be split as (1) *post-training* pruning methods, which start from an accurate dense baseline, and remove weights, followed by fine-tuning; and (2) *sparse training* methods, which perform weight removal during the training process itself. (Other categories such as *data-free* pruning methods [39, 54] exist, but they are beyond our scope.) We focus on *sparse training*, although we will also compare against state-of-the-art post-training methods.

Arguably, the most popular metric for weight removal is *weight magnitude* [24, 26, 62]. Better-performing approaches exist, such as second-order metrics [38, 27, 14, 52], or Bayesian approaches [46], but they tend to have higher computational and implementation cost.

The general goal of *sparse training* methods is to perform both the forward (inference) pass *and the backpropagation* pass over a sparse support, leading to computational gains during the training process as well. One of the first approaches to maintain sparsity throughout training was Deep Rewiring [4], where SGD steps applied to positive weights are augmented with random walks in parameter space, followed by inactivating negative weights. To maintain sparsity throughout training, randomly chosen inactive connections are re-introduced in the "growth" phase. Sparse Evolutionary Training (SET) [44] introduces a non-uniform sparsity distribution across layers, which scales with the number of input and output channels, and trains sparse networks by pruning weights with smallest magnitude and re-introducing some weights randomly. RigL [16] prunes weights at random after a warm-up period, and then periodically performs weight re-introduction using a combination of connectivity- and gradient-based statistics, which require periodically evaluating full gradients. RigL can lead to state-of-the-art accuracy results even compared to post-training methods; however, to achieve high accuracy it requires significant additional data passes (e.g. 5x) relative to the dense baseline. Top-KAST [32] alleviated the drawback of periodically having to evaluate dense gradients by updating the sparsity masks using gradients *of reduced sparsity* relative to the weight sparsity. The latter two methods set the state-of-the-art for sparse training: when executing for the same number of epochs as the dense baseline, they provide computational reductions the order of 2x, while the accuracy of the resulting sparse models is lower than that of leading post-training methods, executed at the same sparsity levels. To our knowledge, none of these methods have convergence guarantees.

Another approach towards faster training is *training sparse networks from scratch*. The masks are updated by continuously pruning and re-introducing weights. For example, [40] uses magnitude pruning after applying SGD on the dense network, whereas [13] update the masks by re-introducing weights with the highest gradient momentum. STR [37] learns a separate pruning threshold for each layer and allows sparsity both during forward and backward passes; however, the desired sparsity can not be explicitly imposed, and the network has low sparsity for a large portion of training. These methods can lead to only limited computational gains, since they either require dense gradients, or the sparsity level cannot be imposed. By comparison, our method provides models of similar or better accuracy at the same sparsity, with computational reductions. We also obtain dense models that match the baseline accuracy, with a fraction of the baseline FLOPs.

The idea of alternating sparse and dense training phases has been examined before in the context of neural networks, but with the goal of using temporary sparsification as a regularizer. Specifically,

*Dense-Sparse-Dense (DSD)* [25] proposes to first *train a dense model to full accuracy*; this model is then sparsified via magnitude; next, optimization is performed over the sparse support, followed by an additional optimization phase over the full dense support. Thus, this process is used as a regularization mechanism for the dense model, which results in relatively small, but consistent accuracy improvements relative to the original dense model. In [33], the authors propose a similar approach to DSD, but alternate sparse phases during the regular training process. The resulting process is similar to AC/DC, but, importantly, the goal of their procedure is to return a *more accurate dense model*. (Please see their Algorithm 1.) For this, the authors use relatively low sparsity levels, and gradually increase sparsity during optimization; they observe accuracy improvements for the resulting dense models, at the cost of increasing the total number of epochs of training. By contrast, our focus is on obtaining accurate *sparse* models, while reducing computational cost, and executing the dense training recipe. We execute at higher sparsity levels, and on larger-scale datasets and models. In addition, we also show that the method works for other sparsity patterns, e.g. the 2:4 semi-structured pattern [43].

More broadly, the Lottery Ticket Hypothesis (LTH) [19] states that sparse networks can be trained in isolation *from scratch* to the same performance as a post-training pruning baseline, by starting from the "right" weight and sparsity mask initializations, optimizing only over this sparse support. However, initializations usually require the availability of the fully-trained dense model, falling under *post-training* methods. There is still active research on replicating these intriguing findings to large-scale models and datasets [21, 20]. Previous work [21, 62] have studied progressive sparsification during regular training, which may also achieve training time speed-up, after a sufficient sparsity level has been achieved. However, AC/DC generally achieves a better trade-off between validation accuracy and training time speed-up, compared to these methods.

Parallel work by [45] investigates a related approach, but focusing on low-rank decompositions for Transformer models. Both their analytical approach and their application domain are different to the ones of the current work.

## 3 Alternating Compressed / DeCompressed (AC/DC) Training

### 3.1 Background and Assumptions

Obtaining *sparse* solutions to optimization problems is a problem of interest in several areas [7, 6, 17], where the goal is to minimize a function $f : \mathbb{R}^N \to \mathbb{R}$ under sparsity constraints:

$$\min_{\theta \in \mathbb{R}^N} f(\theta) \quad \text{s.t.} \quad \|\theta\|_0 \leq k. \tag{1}$$

For the case of $\ell_2$ regression, $f(\theta) = \|b - A\theta\|_2^2$, a solution has been provided by Blumensath and Davies [6], known as the *Iterative Hard Thresholding (IHT)* algorithm, and subsequent work [17, 18, 58] provided theoretical guarantees for the linear operators used in compressed sensing. The idea consists of alternating gradient descent (GD) steps and applications of a thresholding operator to ensure the $\ell_0$ constraint is satisfied. More precisely, $T_k$ is defined as the "top-k" operator, which keeps the largest $k$ entries of a vector $\theta$ in absolute value, and replaces the rest with $0$. The IHT update at step $t + 1$ has the following form:

$$\theta_{t+1} = T_k(\theta_t - \eta \nabla f(\theta_t)). \tag{2}$$

Most convergence results for IHT assume deterministic gradient descent steps. For DNNs, stochastic methods are preferred, so we describe and analyze a stochastic version of IHT.

**Stochastic IHT.** We consider functions $f : \mathbb{R}^N \to \mathbb{R}$, for which we can compute stochastic gradients $g_\theta$, which are unbiased estimators of the true gradient $\nabla f(\theta)$. Define the *stochastic* IHT update as:

$$\theta_{t+1} = T_k(\theta_t - \eta g_{\theta_t}). \tag{3}$$

This formulation covers the practical case where the stochastic gradient $g_\theta$ corresponds to a *mini-batch* stochastic gradient. Indeed, as in practice $f$ takes the form $f(\theta) = \frac{1}{m} \sum_{i=1}^{m} f(\theta; x_i)$, where $S = \{x_1, \ldots, x_m\}$ are data samples, the stochastic gradients obtained via backpropagation take the form $\frac{1}{|B|} \sum_{i \in B} \nabla f(\theta; x_i)$, where $B$ is a sampled mini-batch. We aim to prove strong convergence bounds for stochastic IHT, under common assumptions that arise in the context of training DNNs.

**Analytical Assumptions.** Formally, our analysis uses the following assumptions on $f$.

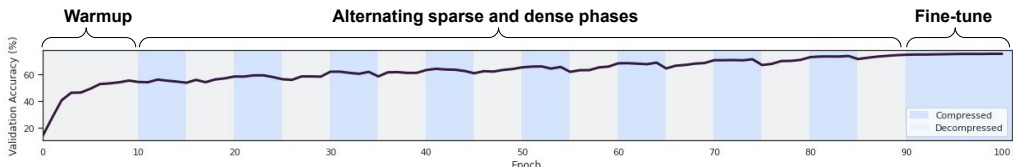

Figure 1: The AC/DC training process. After a short warmup we alternatively prune to maximum sparsity and restore the pruned weights. The plot shows the sparsity and validation accuracy throughout the process for a sample run on ResNet50/ImageNet at 90% sparsity.

1) Unbiased gradients with variance $\sigma$: $\mathbb{E}[g_\theta|\theta] = \nabla f(\theta)$, and $\mathbb{E}[\|g_\theta - \nabla f(\theta)\|^2] \leq \sigma^2$.

2) Existence of a $k^*$-sparse minimizer $\theta^*$: $\exists \theta^* \in \arg\min_\theta f(\theta)$, s.t. $\|\theta^*\|_0 \leq k^*$.

3) For $\beta > 0$, the $\beta$-smoothness condition when restricted to $t$ coordinates $((t, \beta)$-smoothness$)$:

$$f(\theta + \delta) \leq f(\theta) + \nabla f(\theta)^\top \delta + \frac{\beta}{2}\|\delta\|^2, \text{ for all } \theta, \delta \text{ s.t. } \|\delta\|_0 \leq t. \tag{4}$$

4) For $\alpha > 0$ and number of indices $r$, the $r$-concentrated Polyak-Łojasiewicz $((r, \alpha)$-CPL$)$ condition:

$$\|T_r(\nabla f(\theta))\| \geq \frac{\alpha}{2}(f(\theta) - f(\theta^*)), \text{ for all } \theta. \tag{5}$$

The first assumption is standard in stochastic optimization, while the existence of very sparse minimizers is a known property in over-parametrized DNNs [19], and is the very premise of our study. Smoothness is also a standard assumption, e.g. [40]—we only require it along *sparse* directions, which is a strictly weaker assumption. The more interesting requirement for our convergence proof is the $(r, \alpha)$-CPL condition in Equation (5), which we now discuss in detail.

The standard Polyak-Łojasiewicz (PL) condition [34] is common in non-convex optimization, and versions of it are essential in the analysis of DNN training [41, 2]. Its standard form states that small gradient norm, i.e. approximate stationarity, implies closeness to optimum in function value. We require a slightly stronger version, in terms of the norm of the gradient contributed by its largest coordinates in absolute value. This restriction appears necessary for the success of IHT methods, as the sparsity enforced by the truncation step automatically reduces the progress ensured by a gradient step to an amount proportional to the norm of the top-$k$ gradient entries. This strengthening of the PL condition is supported both theoretically, by the mean-field view, which argues that gradients are sub-gaussian [50], and by empirical validations of this behaviour [1, 51].

We are now ready to state our main analytical result.

**Theorem 1.** *Let $f : \mathbb{R}^N \to \mathbb{R}$ be a function with a $k^*$-sparse minimizer $\theta^*$. Let $\beta > \alpha > 0$ be parameters, let $k = C \cdot k^* \cdot (\beta/\alpha)^2$ for some appropriately chosen constant $C$, and suppose that $f$ is $(2k + 3k^*, \beta)$-smooth and $(k^*, \alpha)$-CPL. For initial parameters $\theta_0$ and precision $\epsilon > 0$, given access to stochastic gradients with variance $\sigma$, stochastic IHT (3) converges in $O\left(\frac{\beta}{\alpha} \cdot \ln \frac{f(\theta_0) - f(\theta^*)}{\epsilon}\right)$ iterations to a point $\theta$ with $\|\theta\|_0 \leq k$, such that*

$$\mathbb{E}\left[f(\theta) - f(\theta^*)\right] \leq \epsilon + \frac{16\sigma^2}{\alpha}.$$

Assuming a fixed objective function $f$ and tolerance $\epsilon$, we can obtain lower loss and faster running time by either increasing the support $k$ demanded from our approximate minimizer $\theta$ relative to the optimal $k^*$, or by reducing the gradient variance. We provide a complete proof of this result in the Supplementary Material. Our analysis approach also works in the absence of the CPL condition (Theorem 3), in which case we prove that a version of the algorithm can find sparse nearly-stationary points. As a bonus, we also simplify existing analyses for IHT and extend them to the stochastic case (Theorem 2). Another interpretation of our results is in showing that, under our assumptions, *error feedback* [40] is not necessary for recovering good sparse minimizers; this has practical implications, as it allows us to perform fully-sparse back-propagation in sparse optimization phases. Next, we discuss our practical implementation, and its connection to these theoretical results.

---

**Algorithm 1** Alternating Compressed/Decompressed (AC/DC) Training

---

**Require:** Weights $\theta \in \mathbb{R}^N$, data $S$, sparsity $k$, compression phases $\mathcal{C}$, decompression phases $\mathcal{D}$
  1: Train the weights $\theta$ for $\Delta_w$ epochs                                                       ▷ Warm-up phase
  2: **while** epoch $\leq$ max epochs **do**
  3:   **if** entered a compression phase **then**
  4:     $\theta \leftarrow T_k(\theta, k)$                                        ▷ apply compression (top-k) operator on weights
  5:     $m \leftarrow \mathbb{1}[\theta_i \neq 0]$                                                          ▷ create masks
  6:   **end if**
  7:   **if** entered a decompression phase **then**
  8:     $m \leftarrow \mathbb{1}_N$                                                                ▷ reset all masks
  9:   **end if**
 10:   $\theta \leftarrow \theta \odot m$                            ▷ apply the masks (ensure sparsity for compression phases)
 11:   $\tilde{\theta} \leftarrow \{\theta_i | m_i \neq 0, 1 \leq i \leq N\}$                              ▷ get the support for the gradients
 12:   **for** x mini-batch in $S$ **do**
 13:     $\theta \leftarrow \theta - \eta \nabla_{\tilde{\theta}} f(\theta; x)$                               ▷ optimize the active weights
 14:   **end for**
 15:   epoch $\leftarrow$ epoch $+1$
 16: **end while**
 17: **return** $\theta$

---

## 3.2 AC/DC: Applying IHT to Deep Neural Networks

AC/DC starts from a standard DNN training flow, using standard optimizers such as SGD with momentum [48] or Adam [35], and it preserves all standard training hyper-parameters. It will only periodically modify the *support* for optimization. Please see Algorithm 1 for pseudocode.

We partition the set of training epochs into *compressed* epochs $\mathcal{C}$, and *decompressed* epochs $\mathcal{D}$. We begin with a *dense warm-up* period of $\Delta_w$ consecutive epochs, during which regular dense (decompressed) training is performed. We then start alternating *compressed optimization* phases of length $\Delta_c$ epochs each, with *decompressed (regular) optimization* phases of length $\Delta_d$ epochs each. The process completes on a compressed fine-tuning phase, returning an accurate sparse model. Alternatively, if our goal is to return a dense model matching the baseline accuracy, we take the best dense checkpoint obtained during alternation, and fine-tune it over the entire support. In practice, we noticed that allowing a longer final decompressed phase of length $\Delta_D > \Delta_d$ improves the performance of the dense model, by allowing it to better recover the baseline accuracy after fine-tuning. Please see Figure 1 for an illustration of the schedule.

In our experiments, we focus on the case where the compression operation is unstructured or semi-structured pruning. In this case, at the beginning of each sparse optimization phase, we apply the top-k operator across all of the network weights to obtain a mask $M$ over the weights $\theta$. The top-k operator is applied globally across all of the network weights, and will represent the sparse support over which optimization will be performed for the rest of the current sparse phase. At the end of the sparse phase, the mask $M$ is reset to all-1s, so that the subsequent dense phase will optimize over the full dense support. Furthermore, once all weights are re-introduced, it is beneficial to reset to $0$ the gradient momentum term of the optimizer; this is particularly useful for the weights that were previously pruned, which would otherwise have stale versions of gradients.

**Discussion**. Moving from IHT to a robust implementation in the context of DNNs required some adjustments. First, each *decompressed phase* can be directly mapped to a *deterministic/stochastic IHT* step, where, instead of a single gradient step in between consecutive truncations of the support, we perform several stochastic steps. These additional steps improve the accuracy of the method in practice, and we can bound their influence in theory as well, although they do not necessarily provide better bounds. This leaves open the interpretation of the *compressed phases*: for this, notice that the core of the proof for Theorem 1 is in showing that a single IHT step significantly decreases the expected value of the objective; using a similar argument, we can prove that additional optimization steps over the sparse support can only improve convergence. Additionally, we show convergence for a variant of IHT closely following AC/DC (please see Corollary 1 in the Supplementary Material), but the bounds do not improve over Theorem 1. However, this additional result confirms that the good experimental results obtained with AC/DC are theoretically motivated.

# 4 Experimental Validation

**Goals and Setup.** We tested AC/DC on image classification tasks (CIFAR-100 [36] and ImageNet [49]) and on language modelling tasks [42] using the Transformer-XL model [10]. The goal is to examine the *validation accuracy* of the resulting sparse and dense models, versus the induced sparsity, as well as the number of FLOPs used for training and inference, relative to other sparse training methods. Additionally, we compare to state-of-the-art post-training pruning methods [52]. We also examine prediction differences between the sparse and dense models. We use PyTorch [47] for our implementation, Weights & Biases [5] for experimental tracking, and NVIDIA GPUs for training. All reported image classification experiments were performed in triplicate by varying the random seed; we report mean and standard deviation. Due to computational limitations, the language modelling experiments were conducted in a single run.

**ImageNet Experiments.** On the ImageNet dataset [49], we test AC/DC on ResNet50 [28] and MobileNetV1 [30]. In all reported results, the models were trained for a fixed number of 100 epochs, using SGD with momentum. We use a cosine learning rate scheduler and training hyper-parameters following [37], but without label smoothing. The models were trained and evaluated using mixed precision (FP16). On a small subset of experiments, we noticed differences in accuracy of up to 0.2-0.3% between AC/DC trained with full or mixed precision. However, the differences in evaluating the models with FP32 or FP16 are negligible (less than 0.05%). Our dense ResNet50 baseline has 76.84% validation accuracy. Unless otherwise specified, weights are pruned globally, based on their magnitude and in a single step. Similar to previous work, we did not prune biases, nor the Batch Normalization parameters. The sparsity level is computed with respect to all the parameters, except the biases and Batch Normalization parameters and this is consistent with previous work [16, 52].

For all results, the AC/DC training schedule starts with a "warm-up" phase of dense training for 10 epochs, after which we alternate between compression and de-compression every 5 epochs, until the last dense and sparse phase. It is beneficial to allow these last two "fine-tuning" phases to run longer: the last decompression phase runs for 10 epochs, whereas the final 15 epochs are the compression fine-tuning phase. We reset SGD momentum at the beginning of every decompression phase. In total, we have an equal number of epochs of dense and sparse training; see Figure (2a) for an illustration. We use exactly the same setup for both ResNet50 and MobileNetV1 models, which resulted in high-quality sparse models. To recover a dense model with baseline accuracy using AC/DC, we finetune the best dense checkpoint obtained during training; practically, this replaces the last *sparse* fine-tuning phase with a phase where the *dense* model is fine-tuned instead.

Table 1: ResNet50/ImageNet, medium sparsity results.

| Method | Sparsity (%) | Top-1 Acc. (%) | GFLOPs Inference | EFLOPs Train |
|---|---|---|---|---|
| Dense | 0 | 76.84 | 8.2 | 3.14 |
| **AC/DC** | 80 | $76.3 \pm 0.1$ | 0.29× | 0.65× |
| RigL$_{1\times}$ | 80 | $74.6 \pm 0.06$ | 0.23× | 0.23× |
| RigL$_{1\times}$(ERK) | 80 | $75.1 \pm 0.05$ | 0.42× | 0.42× |
| Top-KAST | 80 fwd, 50 bwd | 75.03 | 0.23× | 0.32× |
| STR | 79.55 | 76.19 | 0.19× | - |
| **WoodFisher** | 80 | **76.76** | 0.25× | - |
| **AC/DC** | 90 | $75.03 \pm 0.1$ | 0.18× | 0.58× |
| RigL$_{1\times}$ | 90 | $72.0 \pm 0.05$ | 0.13× | 0.13× |
| RigL$_{1\times}$ (ERK) | 90 | $73.0 \pm 0.04$ | 0.24× | 0.25× |
| Top-KAST | 90 fwd, 80 bwd | 74.76 | 0.13× | 0.16× |
| STR | 90.23 | 74.31 | 0.08× | - |
| **WoodFisher** | 90 | **75.21** | 0.15× | - |

Table 2: ResNet50/ImageNet, high sparsity results.

| Method | Sparsity (%) | Top-1 Acc. (%) | GFLOPs Inference | EFLOPs Train |
|---|---|---|---|---|
| Dense | 0 | 76.84 | 8.2 | 3.14 |
| **AC/DC** | 95 | **$73.14 \pm 0.2$** | 0.11× | 0.53× |
| RigL$_{1\times}$ | 95 | $67.5 \pm 0.1$ | 0.08× | 0.08× |
| RigL$_{1\times}$ (ERK) | 95 | $69.7 \pm 0.17$ | 0.12× | 0.13× |
| Top-KAST | 95 fwd, 50 bwd | 71.96 | 0.08× | 0.22× |
| STR | 94.8 | 70.97 | 0.04× | - |
| WoodFisher | 95 | 72.12 | 0.09× | - |
| **AC/DC** | 98 | **$68.44 \pm 0.09$** | 0.06× | 0.46× |
| Top-KAST | 98 fwd, 90 bwd | 67.06 | 0.05× | 0.08× |
| STR | 97.78 | 62.84 | 0.02× | - |
| WoodFisher | 98 | 65.55 | 0.05× | - |

**ResNet50 Results.** Tables 1 & 2 contain the validation accuracy results across medium and high global sparsity levels, as well as inference and training FLOPs. Overall, AC/DC achieves higher validation accuracy than any of the state-of-the-art sparse training methods, when using the same number of epochs. At the same time, due to dense training phases, AC/DC has higher FLOP requirements relative to RigL or Top-KAST at the same sparsity. At medium sparsities (80% and 90%), AC/DC sparse models are slightly less accurate than the state-of-the-art post-training methods (e.g. WoodFisher), by small margins. The situation is reversed at higher sparsities, where AC/DC produces more accurate models: the gap to the second-best methods (WoodFisher / Top-KAST) is of more than 1% at 95% and 98% sparsity.

Of the existing sparse training methods, Top-KAST is closest in terms of validation accuracy to our sparse model, at 90% sparsity. However, Top-KAST does not prune the first and last layers,

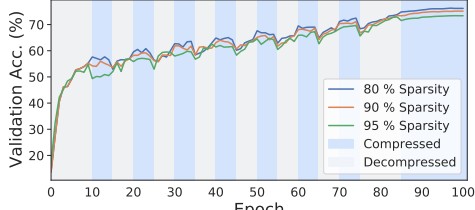
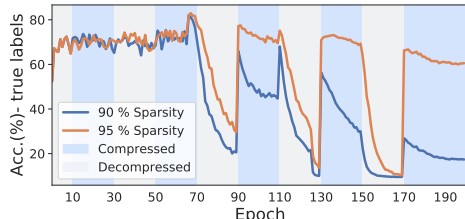

(a) Sparsity pattern and validation accuracy vs. number of epochs (ResNet50/ImageNet).

(b) Percentage of samples with corrupted training labels classified to their *true* class (ResNet20/CIFAR10).

Figure 2: Accuracy vs. sparsity during training, for the ResNet50/ImageNet experiment (left) and accuracy on the corrupted samples for ResNet20/CIFAR10, w.r.t. the *true* class (right).

whereas the results in the tables do not restrict the sparsity pattern. For fairness, we executed AC/DC using the same layer-wise sparsity distribution as Top-KAST, for both uniform and global magnitude pruning. For $90\%$ global pruning, results for AC/DC improved; the best sparse model reached $75.64\%$ validation accuracy ($0.6\%$ increase over Table 1), while the best dense model had $76.85\%$ after fine-tuning. For uniform sparsity, our results were very similar: $75.04\%$ validation accuracy for the sparse model and $76.43\%$ - for the fine-tuned dense model. We also note that Top-KAST has better results at $98\%$ when increasing the number of training epochs 2 times, and considerably fewer training FLOPs (e.g. $15\%$ of the dense FLOPs). For fairness, we compared against all methods on a fixed number of 100 training epochs and we additionally trained AC/DC at high sparsity without pruning the first and last layers. Our results improved to $74.16\%$ accuracy for $95\%$ sparsity, and $71.27\%$ for $98\%$ sparsity, both surpassing Top-KAST with prolonged training. We provide a more detailed comparison in the Supplementary Material, which also contains results on CIFAR-100.

An advantage of AC/DC is that it provides *both* sparse and dense models at cost *below* that of a single dense training run. For medium sparsity, the accuracy of the dense-finetuned model is very close to the dense baseline. Concretely, at 90% sparsity, with 58% of the total (theoretical) baseline training FLOPs, we obtain a *sparse* model which is close to state of the art; in addition, by fine-tuning the best dense model, we obtain a dense model with $76.56\%$ (average) validation accuracy. The whole process takes at most 73% of the baseline training FLOPs. In general, for 80% and 90% target sparsity, the dense models derived from AC/DC are able to recover the baseline accuracy, after finetuning, defined by replacing the final compression phase with regular dense training. The complete results are presented in the Supplementary Material, in Table 6.

The sparsity distribution over layers does not change dramatically during training; yet, the dynamic of the masks has an important impact on the performance of AC/DC. Specifically, we observed that masks update over time, although the change between consecutive sparse masks decreases. Furthermore, a small percentage of the weights remain fixed at $0$ even during dense training, which is explained by filters that are pruned away during the compressed phases. Please see the Supplementary Material for additional results and analysis.

We additionally compare AC/DC with Top-KAST and RigL, in terms of the validation accuracy achieved depending on the number of training FLOPs. We report results at uniform sparsity, which ensures that the inference FLOPs will be the same for all methods considered. For AC/DC and Top-KAST, the first and last layers are kept dense, whereas for RigL, only the first layer is kept dense; however, this has a negligible impact on the number of FLOPs. Additionally, we experiment with extending the number of training iterations for AC/DC at 90% and 95% sparsity two times, similarly to Top-KAST and RigL which also provide experiments for extended training. The comparison between AC/DC, Top-KAST and RigL presented in Figure 3 shows that AC/DC is similar or surpasses Top-KAST 2x at 90% and 95% sparsity, and RigL 5x at 95% sparsity both in terms of training FLOPs and validation accuracy. Moreover, we highlight that extending the number of training iterations two times results in AC/DC models with uniform sparsity that surpass all existing methods at both 90% and 95% sparsity; namely, we obtain 76.1% and 74.3% validation accuracy with 90% and 95% uniform sparsity, respectively.

Compared to purely sparse training methods, such as Top-KAST or RigL, AC/DC requires dense training phases. The length of the dense phases can be decreased, with a small impact on the accuracy of the sparse model. Specifically, we use dense phases of two instead of five epochs in length, and we

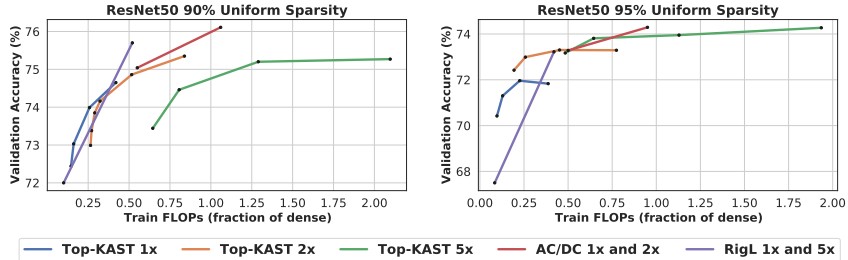

Figure 3: Training FLOPs vs validation accuracy for AC/DC, RigL and Top-KAST, with uniform sparsity, at 90% and 95% sparsity levels. (ResNet50/ImageNet).

no longer extend the final decompressed phase prior to the finetuning phase. For 90% global sparsity, this resulted in 74.6% validation accuracy for the sparse model, using 44% of the baseline FLOPs. Similarly, for uniform sparsity, we obtain 74.7% accuracy on the 90% sparse model, with 40% of the baseline FLOPs; this value can be further improved to 75.8% validation accuracy when extending two times the number of training iterations. Furthermore, at 95% uniform sparsity, we reach 72.8% accuracy with 35% of the baseline training FLOPs.

**MobileNet Results.** We perform the same experiment, using exactly the same setup, on the MobileNetV1 architecture [30], which is compact and thus harder to compress. On a training budget of 100 epochs, our method finds sparse models with higher Top-1 validation accuracy than existing sparse- and post-training methods, on both 75% and 90% sparsity levels (Table 3). Importantly, AC/DC uses exactly the same hyper-parameters used for training the dense baseline [37]. Similar to ResNet50, at 75% sparsity, the *dense-finetuned model* recovers the baseline performance, while for 90% it is less than 1% below the baseline. The only method which obtains higher accuracy for the same sparsity is the version of RigL [16] which executes for 5x more training epochs than the dense baseline. However, this version also uses more computation than the dense model. We limit ourselves to a fixed number of 100 epochs, the same used to train the dense baseline, which would allow for savings in training time. Moreover, RigL does not prune the first layer and the depth-wise convolutions, whereas for the results reported we do not impose any sparsity restrictions. Overall, we found that keeping these layers dense improved our results on 90% sparsity by almost 0.5%. Then, our results are quite close to RigL$_{2\times}$, with half the training epochs, and less training FLOPs. We provide a more detailed comparison in the Supplementary Material.

Table 3: MobileNetV1/ImageNet sparsity results

| Method | Sparsity (%) | Top-1 Acc. (%) | GFLOPs Inference | EFLOPs Train |
|---|---|---|---|---|
| Dense | 0 | 71.78 | 1.1 | 0.44 |
| **AC/DC** | 75 | **70.3 ± 0.07** | 0.34× | 0.64× |
| RigL$_{1\times}$ (ERK) | 75 | 68.39 | 0.52× | 0.53× |
| STR | 75.28 | 68.35 | 0.18× | - |
| WoodFisher | 75.28 | 70.09 | 0.28× | - |
| **AC/DC** | 90 | **66.08 ± 0.09** | 0.18× | 0.56× |
| RigL$_{1\times}$ (ERK) | 90 | 63.58 | 0.27× | 0.29× |
| STR | 89.01 | 62.1 | 0.07× | - |
| WoodFisher | 89 | 63.87 | - | - |

Table 4: Transformer-XL/WikiText sparsity results

| Method | Sparsity (%) | Perplexity Sparse | Perplexity Dense | Perplexity Finetuned Dense |
|---|---|---|---|---|
| Dense | 0 | - | 18.95 | - |
| AC/DC | 80 | 20.65 | 20.24 | 19.54 |
| AC/DC | 80, 50 embed. | 20.83 | 20.25 | 19.68 |
| **Top-KAST** | 80, 0 bwd | **19.8** | - | - |
| Top-KAST | 80, 60 bwd | 21.3 | - | - |
| **AC/DC** | 90 | **22.32** | 21.0 | 20.28 |
| **AC/DC** | 90, 50 embed. | **22.84** | 21.34 | 20.41 |
| Top-KAST | 90, 80 bwd | 25.1 | - | - |

**Semi-structured Sparsity.** We also experiment with the recent 2:4 sparsity pattern (2 weights out of each block of 4 are zero) proposed by NVIDIA, which ensures inference speedups on the Ampere architecture. Recently, [43] showed that accuracy can be preserved under this pattern, by re-doing the entire training flow. Also, [61] proposed more general N:M structures, together with a method for training such sparse models from scratch. We applied AC/DC to the 2:4 pattern, performing training from scratch and obtained sparse models with $76.64\% \pm 0.05$ validation accuracy, i.e. slightly below the baseline. Furthermore, the dense-finetuned model fully recovers the baseline performance (76.85% accuracy). We additionally experiment with using AC/DC with global pruning at 50%; in this case we obtain sparse models that slightly improve the baseline accuracy to 77.05%. This confirms our intuition that AC/DC can act as a regularizer, similarly to [25].

**Language Modeling.** Next, we apply AC/DC to compressing NLP models. We use Transformer-XL [10], on the WikiText-103 dataset [42], with the standard model configuration with 18 layers and 285M parameters, trained using the Lamb optimizer [57] and standard hyper-parameters, which we

describe in the Supplementary Material. The same Transformer-XL model trained on WikiText-103 was used in Top-KAST [32], which allows a direct comparison. Similar to Top-KAST, we did not prune the embedding layers, as this greatly affects the quality, without reducing computational cost. (For completeness, we do provide results when embeddings are pruned to 50% sparsity.) Our sparse training configuration consists in starting with a dense warm-up phase of 5 epochs, followed by alternating between compression and decompression phases every 3 epochs; we follow with a longer decompression phase between epochs 33-39, and end with a compression phase between epochs 40-48. The results are shown in Table 4. Relative to Top-KAST, our approach provides significantly improved test perplexity at 90% sparsity, as well as better results at 80% sparsity with sparse back-propagation. The results confirm that AC/DC is scalable and extensible. We note that our hyper-parameter tuning for this experiment was minimal.

**Output Analysis.** Finally, we probe the accuracy difference between the sparse and dense-finetuned models. We first examineed *sample-level agreement* between sparse and dense-finetuned pairs produced by AC/DC, relative to model pairs produced by gradual magnitude pruning (GMP). Co-trained model pairs consistently agree on more samples relative to GMP: for example, on the 80%-pruned ResNet50 model, the AC/DC model pair agrees on the Top-1 classification of 90% of validation samples, whereas the GMP models agree on 86% of the samples. The differences are better seen in terms of validation error (10% versus 14%), which indicate that the dense baseline and GMP model disagree on 40% more samples compared to the AC/DC models. A similar trend holds for the *cross-entropy* between model outputs. This is a potentially useful side-effect of the method; for example, in constrained environments where sparse models are needed, it is important to estimate their similarity to the dense ones.

Second, we analyze differences in "memorization" capacity [60] between dense and sparse models. For this, we apply AC/DC to ResNet20 trained on a variant of CIFAR-10 where a subset of 1000 samples have randomly corrupted class labels, and examine the accuracy on these samples during training. We consider 90% and 95% sparsity AC/DC runs. Figure 2b shows the results, when the accuracy for each sample is measured with respect to the *true, un-corrupted* label. During early training and during *sparse phases*, the network tends to classify corrupted samples to their *true class*, "ignoring" label corruption. However, as training progresses, due to dense training phases and lower learning rate, networks tend to "memorize" these samples, assigning them to their corrupted class. This phenomenon is even more prevalent at 95% sparsity, where the network is less capable of memorization. We discuss this finding in more detail in the Supplementary Material.

**Practical Speedups.** One remaining question regards the potential of sparsity to provide real-world speedups. While this is an active research area, e.g. [15], we partially address this concern in the Supplementary Material, by showing inference speedups for our models on a CPU inference platform supporting unstructured sparsity [12]: for example, our 90% sparse ResNet50 model provides 1.75x speedup for real-time inference (batch-size 1) on a resource-constrained processor with 4 cores, and 2.75x speedup on 16 cores at batch size 64, versus the dense model.

# 5 Conclusion, Limitations, and Future Work

We introduced AC/DC—a method for co-training sparse and dense models, with theoretical guarantees. Experimental results show that AC/DC improves upon the accuracy of previous sparse training methods, and obtains state-of-the-art results at high sparsities. Importantly, we recover near-baseline performance for dense models and do not require extensive hyper-parameter tuning. We also show that AC/DC has potential for real-world speed-ups in inference and training, with the appropriate software and hardware support. The method has the advantage of returning both an accurate standard model, and a compressed one. Our model output analysis confirms the intuition that sparse training phases act as a regularizer, preventing the (dense) model from memorizing corrupted samples. At the same time, they prevent the memorization of *hard samples*, which can affect accuracy.

The main limitations of AC/DC are its reliance on dense training phases, which limits the achievable training speedup, and the need for tuning the length and frequency of sparse/dense phases. We believe the latter issue can be addressed with more experimentation (we show some preliminary results in Section 4 and Appendix B.1); however, both the theoretical results and the output analysis suggest that dense phases may be *necessary* for good accuracy. We plan to further investigate this in future work, together with applying AC/DC to other compression methods, such as quantization, as well as leveraging sparse training on hardware that could efficiently support it, such as Graphcore IPUs [23].

## Acknowledgments and Disclosure of Funding

This project has received funding from the European Research Council (ERC) under the European Union's Horizon 2020 research and innovation programme (grant agreement No 805223 ScaleML), and a CNRS PEPS grant. This research was supported by the Scientific Service Units (SSU) of IST Austria through resources provided by Scientific Computing (SciComp). We would also like to thank Christoph Lampert for his feedback on an earlier version of this work, as well as for providing hardware for the Transformer-XL experiments.

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
