# Appendices

## Contents

## A Convergence Proofs

In this section we provide the convergence analysis for our algorithms. We prove Theorem 1 and show as a corollary that under reasonable assumptions our implementation of AC/DC converges to a sparse minimizer.

## A.1 Overview

We use the notation and assumptions defined in Section 3.1. As all of our analyses revolve around bounding the progress made in a single iteration, to simplify notation we will generally use $\theta$ to denote the current iterate, and $\theta'$ to denote the iterate obtained after the IHT update:

$$\theta' = T_k(\theta - \eta g_\theta).$$

Additionally, we let $S, S', S^* \subseteq [N]$ to denote the support of $\theta$, $\theta'$ and $\theta^*$ respectively, where $\theta^*$ is the promised $k^*$-sparse minimizer. Given an arbitrary vector $x$, we use $\text{supp}(x)$ to denote its support, i.e. the set of coordinates where $x$ is nonzero. We may also refer to the minimizing value of $f$ as $f^* = f(\theta^*)$.

Before providing more intuition for the analysis, we give the main theorem statements.

### A.1.1 Stochastic IHT for Functions with Concentrated PL Condition

We restate the Theorem from Section 3.1.

**Theorem 1.** *Let $f : \mathbb{R}^N \to \mathbb{R}$ be a function with a $k^*$-sparse minimizer $\theta^*$. Let $\beta > \alpha > 0$ be parameters, let $k = C \cdot k^* \cdot (\beta/\alpha)^2$ for some appropriately chosen constant $C$, and suppose that $f$ is $(2k + 3k^*, \beta)$-smooth and $(k^*, \alpha)$-CPL. For initial parameters $\theta_0$ and precision $\epsilon > 0$, given access to stochastic gradients with variance $\sigma$, stochastic IHT (3) converges in $O\left(\frac{\beta}{\alpha} \cdot \ln \frac{f(\theta_0) - f(\theta^*)}{\epsilon}\right)$ iterations to a point $\theta$ with $\|\theta\|_0 \leq k$, such that*

$$\mathbb{E}\left[f(\theta) - f(\theta^*)\right] \leq \epsilon + \frac{16\sigma^2}{\alpha}.$$

Additionally, we give a corollary that justifies our implementation of AC/DC. As opposed to the theoretical stochastic IHT algorithm, AC/DC performs a sequence of several dense SGD steps before applying a single pruning step. We show that even with this change we can provide theoretical convergence bounds, although these bounds can be weaker than the baseline IHT method under our assumptions.

**Corollary 1** (Convergence of AC/DC). *Let $f : \mathbb{R}^N \to \mathbb{R}$ be a function that decomposes as $f(\theta) = \frac{1}{m}\sum_{i=1}^m f_i(\theta)$, and has a $k^*$-sparse minimizer $\theta^*$. Let $\beta > \alpha > 0$ be parameters, let $k = C \cdot k^* \cdot (\beta/\alpha)^2$ for some appropriately chosen constant $C$, suppose that each $f_i$ is $(N, \beta)$-smooth, and L-Lipschitz, and that $f$ is $(k^*, \alpha)$-CPL.*

*Let $\Delta_c$ and $B$ be integers, and let $\{S_1, \ldots, S_B\}$ be a partition of $[m]$ into $B$ subsets of cardinality $O(m/B)$ each. Given $\theta$, let $g_\theta^{(i)} = \frac{1}{|S_i|}\sum_{j \in S_i} \nabla f_j(\theta)$.*

*Suppose we replace the IHT iteration with a dense/sparse phase consisting of*

*1) $\Delta_c$ phases during each of which we perform a full pass over the data by performing the iteration $\theta' = \theta - \eta g_\theta^{(i)}$ for all $i \in [B]$, with an appropriate step size $\eta$;*

*2) a pruning step implemented via an application the truncation operator $T_k$;*

*3) an optional sparse training phase which fully optimizes $f$ over the sparse support.*

*For initial parameters $\theta_0$ and precision $\epsilon > 0$, this algorithm converges in $O\left(\frac{\beta}{\alpha} \cdot \ln \frac{f(\theta_0) - f(\theta^*)}{\epsilon}\right)$ dense/sparse phases to a point $\theta$ with $\|\theta\|_0 \leq k$, such that*

$$f(\theta) - f(\theta^*) \leq \epsilon + O\left(\frac{L^2}{\alpha}\right).$$

To provide more intuition for these results, let us understand how various parameters affect convergence. As we will argue in more detail in Section A.1.4, the main idea behind these algorithms is based on the vanilla IHT algorithm, which consists of alternating full gradient steps with pruning/truncation steps. Pruning to the largest $k$ coordinates in absolute value essentially represents projecting the current weights onto the non-convex set of $k$-sparse vectors, so a natural approach is to simply try to adapt the analysis of projected gradient methods to this setup. The major caveat of this idea is that projecting onto a non-convex set can potentially undo some of the progress made so far, unlike the standard case of projections over convex sets. However, we can argue that if we set the target sparsity $k$ sufficiently large compared to the promised sparsity $k^*$ of the optimum, the pruning step can only increase the function value by a fraction of how much it was decreased by the full gradient step.

Notably, in order to guarantee this property, we require the target sparsity $k$ to be of order $\Omega(k^* \cdot (\beta/\alpha)^2)$, where $\beta/\alpha$ represents the "restricted condition number" of $f$. Hence "well conditioned" functions allow for better sparsity. The number of iterations is same as in vanilla smooth and strongly convex optimization, which only depends on the function $f$, and not on the target sparsity.

The next step is specializing this approach to the non-convex case, under the restricted smoothness and CPL conditions, when only stochastic gradients are available. To handle non-convexity, we show that in fact a weaker property than strong convexity is required, and we can achieve similar results only under these restricted assumptions. More importantly, we can also handle stochastic gradients, which occur naturally in deep learning. In fact, the stochastic variance ends up contributing an extra additive error of $O(\sigma^2/\alpha)$ to our final error bound (see Theorem 1). This can be reduced by decreasing variance, i.e. taking larger batches. Additionally, this term carries a dependence in the CPL parameter $\alpha$ – the larger $\alpha$, the smaller the error. We leave as an open problem whether this dependence on variance can be eliminated for stochastic IHT methods.

Building on this, in Corollary 1 we provide theoretical guarantees for the algorithm we implemented. Notably, while stochastic IHT takes a single stochastic gradient step, followed by pruning, in practice we replace this with a dense training phase, followed by optimization in the sparse support. We show that this does not significantly change the state of affairs. The dense training phase can be thought of as a replacement for a single stochastic gradient step. The difficulty in AC/DC comes from the fact that the total movement of the weights during the dense phase does not constitute an unbiased estimator for the true gradient, as it was previously the case. However, by damping down the learning rate we can see that under reasonable assumptions this total movement does not differ too much from the step we would have made using a single full gradient. The sparse training phase can only help, since our entire analysis is based on proving that the function value decreases in each step. Hence as long as sparse training reduces the function value, it can only improve convergence.

We provide the main proofs for these statements in Section A.2.

### A.1.2 Stochastic IHT for Functions with Restricted Smoothness and Strong Convexity

We also provide a streamlined analysis for stochastic IHT under standard assumptions. Compared to [31] which achieves similar guarantees (in particular both suffer from a blow-up in sparsity that is quadratic in the restricted condition number of $f$) we significantly simplify the analysis and provide guarantees when only stochastic gradients are available. Notably, the quadratic dependence in condition number can be improved to linear, at the expense of a significantly more sophisticated algorithm which requires a longer running time [3].

**Theorem 2.** *Let $f : \mathbb{R}^N \to \mathbb{R}$ be a function with a $k^*$-sparse minimizer $\theta^*$. Let $\beta > \alpha > 0$ be parameters, let $k = C \cdot k^* \cdot (\beta/\alpha)^2$ for some appropriately chosen constant $C$, and suppose that $f$ is $(2k + k^*, \beta)$-smooth and $(k + k^*, \alpha)$-strongly convex in the sense that*

$$f(\theta + \delta) \geq f(\theta) + \nabla f(\theta)^\top \delta + \frac{\alpha}{2}\|\delta\|^2, \text{ for all } \theta, \delta \text{ s. t. } \|\delta\|_0 \leq k + k^*.$$

*For initial parameters $\theta_0$ and precision $\epsilon > 0$, given access to stochastic gradients with variance $\sigma$, IHT (3) converges in $O\left(\frac{\beta}{\alpha} \cdot \ln \frac{\|\theta_0 - \theta^*\|}{\epsilon/\beta + \sigma^2/(\alpha\beta)}\right)$ iterations to a point $\theta$ with $\|\theta\|_0 \leq k$, such that*

$$\mathbb{E}\left[\|\theta - \theta^*\|^2\right] \leq \frac{\epsilon}{\beta} + \frac{2\sigma^2}{\alpha\beta}$$

*and*

$$\mathbb{E}\left[f(\theta) - f(\theta^*)\right] \leq \epsilon + \frac{2\sigma^2}{\alpha}.$$

We prove this statement in Section A.3.

### A.1.3 Finding a Sparse Nearly-Stationary Point

We can further relax the assumptions, and show that IHT can recover sparse iterates that are nearly-stationary (i.e. have small gradient norm). Finding nearly-stationary points is a standard objective in non-convex optimization. However, enforcing the sparsity guarantee is not. Here we show that IHT can further provide stationarity guarantees even with minimal assumptions.

Intuitively, this results suggests that error feedback may not be necessary for converging to stationary points under parameter sparsity.

In this case, we alternate IHT steps with optimization steps over the sparse support, which reduce the norm of the gradient resteicted to the support to some target error $\epsilon$.

**Theorem 3.** *Let $f : \mathbb{R}^N \to \mathbb{R}$. Let $\beta, \epsilon > 0$ be parameters, let $k$ be the target sparsity, and suppose that $f$ is $(2k, \beta)$-smooth. Furthermore, suppose that after each IHT step with a step size $\eta = \beta^{-1}$ and target sparsity $k$, followed by optimizing over the support to gradient norm $\epsilon$, all the obtained iterates satisfy*

$$\|\theta\|_\infty \leq R_\infty.$$

*For initial parameters $\theta_0$ and precision $\epsilon > 0$, given access to stochastic gradients with variance $\sigma$, in $O\left(\beta \left(f(\theta_0) - f(\theta_T)\right) \cdot \min\left\{\frac{1}{\epsilon^2}, \frac{1}{\beta^2 k R_\infty^2}, \frac{1}{\beta^2 \sigma^2}\right\}\right)$ iterations we can obtain an $k$-sparse iterate $\theta$ such that*

$$\|\nabla f(\theta)\|_\infty = O\left(\beta R_\infty \sqrt{k} + \beta\sigma + \epsilon\right).$$

This theorem provides provable guarantees even in absence of properties that bound distance to the optimum via function value (such as restricted strong convexity or the CPL condition. Instead, it uses the reasonable assumption that the $\ell_\infty$ norm of all the sparse iterates witnessed during the optimization process is bounded by a parameter $R_\infty$. In fact, in practice we often notice that weights stay in a small range, so this assumption is well motivated.

Since it can not possibly offer guarantees on global optimality, this theorem instead provides a sparse near-stationary point, in the sense that the $\ell_\infty$ norm of the gradient at the returned point is small. In fact this depends on several parameters, including the target sparsity $k$, smoothness $\beta$ and the promise on $R_\infty$. It is worth mentioning that the $\ell_\infty$ norm of the gradient depends on the norm of the output sparsity, which is independent on the sparsity of any promise, unlike in the previously analyzed cases.

Several other works attempted to compute sparse near-stationary points [40, 45]. As opposed to these we do not require feedback, nor do we sparsify the gradients. Instead, we simply prune the weights after updating them with a dense gradient, following which we optimize in the sparse support until we reach a point whose gradient has only small coordinates within that support.

Since all iterates are in a small $\ell_\infty$ box, we can show that the progress guaranteed by the dense gradient step alone is "mildly" affected by pruning. Hence unless we are already at a near-stationary point in, which case the algorithm can terminate, we decrease the value of $f$ significantly, while only suffering a bit of penalty which is bounded by the parameters of the instance.

We provide full proofs in Section A.4.

### A.1.4 Proof Approach

Let us briefly explain the intuition behind our theoretical analyses. We can view IHT as a version of projected gradient descent, where iterates are projected onto the *non-convex* domain of $k$-sparse vectors.

In general, when the domain is convex, projections do not hurt convergence. This is because under specific assumptions, gradient descent makes progress by provably decreasing the $\ell_2$ distance between the iterate and the optimal solution. As projecting the iterate back onto the convex domain can only improve the distance to the optimum, convergence is unaffected even in those constrained settings.

In our setting, as the set of $k$-sparse vectors is non-convex, the distance to the optimal solution can increase. However, we can show a trade-off between how much this distance increases and the ratio between the target and optimal sparsity $k/k^*$.

**Intuition.** Intuitively, consider a point $\widetilde{\theta}$ obtained by taking a gradient step

$$\widetilde{\theta} = \theta - \eta \nabla f(\theta) . \tag{6}$$

While this step provably decreases the distance to the optimum $\|\widetilde{\theta} - \theta^*\| \ll \|\theta - \theta^*\|$, after applying the projection by moving to the projection $T_k(\widetilde{\theta})$, the distance $\|T_k(\widetilde{\theta}) - \theta^*\|$ may increase again. The key is that this increase can be controlled. For example, the new distance can be bounded via triangle inequality by

$$\|T_k(\widetilde{\theta}) - \theta^*\| \le \|\widetilde{\theta} - T_k(\widetilde{\theta})\| + \|\widetilde{\theta} - \theta^*\| \le 2\|\widetilde{\theta} - \theta^*\| .$$

The last inequality follows from the fact that by definition $T_k(\widetilde{\theta})$ is the closest $k$-sparse vector to $\widetilde{\theta}$ in $\ell_2$ norm, and thus the additional distance payed to move to this projected point is bounded by $\|\widetilde{\theta} - \theta^*\|$. Thus, if the gradient step (6) made sufficient progress, for example $\|\widetilde{\theta} - \theta^*\| \le \frac{1}{3}\|\theta - \theta^*\|$ we can conclude that the additional truncating step does not fully undo progress, as

$$\|T_k(\widetilde{\theta}) - \theta^*\| \le 2\|\widetilde{\theta} - \theta^*\| \le \frac{2}{3}\|\theta - \theta^*\| ,$$

so the iterate still converges to the optimum.

In reality, we can not always guarantee that a single gradient step reduces the distance by a large constant fraction – as a matter of fact this is determined by how well the function $f$ is conditioned. However we can reduce the lost progress that is caused by the truncation step simply by increasing the number of nonzeros of the target solution, i.e. increasing the ratio $k/k^*$.

This is captured by the following crucial lemma which also appears in [31]. A short proof can be found in Section A.5.

**Lemma 1.** *Let $\theta^*$ be a $k^*$-sparse vector, and let $\theta$ be an $n$-sparse vector. Then*

$$\frac{\|T_k(\theta) - \theta\|^2}{n - k} \le \frac{\|\theta^* - \theta\|^2}{n - k^*} .$$

Its usefulness is made obvious in the proof of Theorem 2 which explicitly tracks as a measure of progress the distance between the current iterate and the sparse optimum. This is shown in detail in Section A.3.

The proofs of Theorems 1 and 3 are slightly more complicated, as they track progress in terms of function value rather than distance to the sparse optimum. However, similar arguments based on Lemma 1 still go through. An added benefit is that these analyses show that alternating IHT steps with gradient steps over the sparse support can only help convergence, as these additional steps further decrease error in function value. This fact is important to theoretically justify the performance of the AC/DC algorithm. In the following sections we provide proofs for our theorems.

## A.2  Stochastic IHT for Non-Convex Functions with Concentrated PL Condition

In this section we prove Theorem 1 by showing that the ideas developed before apply to non-convex settings. We analyze IHT for a class of functions that satisfy a special version of the Polyak-Łojasiewicz (PL) condition [34] which is standard in non-convex optimization, and certain versions of it were essential in several works analyzing the convergence of training methods for deep neural networks [41, 2]. Usually this condition says that small gradient norm i.e. approximate stationarity implies closeness to optimum in function value. Here we use the stronger $(r, \alpha)$-CPL condition (see Equation 5), which considers the norm of the gradient contributed by its largest coordinates in absolute value.

We prove strong convergence bounds for functions that satisfy the CPL condition. Compared to the classical Polyak-Łojasiewicz condition, this adds the additional assumption that most of the mass of the gradient is concentrated on a small subset of coordinates. This phenomenon has been witnessed in several instances, and is implicitly used in [40].

Before proceeding with the proof we again provide a few useful lemmas.

**Lemma 2.** *If $f$ is $(\ell, \beta)$-smooth, then for any $\ell$-sparse $\delta$, one has that*

$$f(\theta + \delta) \le f(\theta) + \frac{\beta}{2} \left\| \left( \frac{1}{\beta} \nabla f(\theta) + \delta \right)_{\text{supp}(\delta)} \right\|^2 - \frac{1}{2\beta} \left\| \nabla f(\theta)_{\text{supp}(\delta)} \right\|^2 .$$

*Proof.* Applying smoothness we bound

$$
\begin{aligned}
f(\theta + \delta) &\le f(\theta) + \langle \nabla f(\theta), \delta \rangle + \frac{\beta}{2} \|\delta\|^2 \\
&= f(\theta) + \frac{1}{2\beta} \|\nabla f(\theta)\|^2 + \langle \nabla f(\theta), \delta \rangle + \frac{\beta}{2} \|\delta\|^2 - \frac{1}{2\beta} \|\nabla f(\theta)\|^2 \\
&= f(\theta) + \frac{1}{2} \left\| \frac{1}{\sqrt{\beta}} \nabla f(\theta) + \sqrt{\beta} \delta \right\|^2 - \frac{1}{2\beta} \|\nabla f(\theta)\|^2 \\
&= f(\theta) + \frac{\beta}{2} \left\| \frac{1}{\beta} \nabla f(\theta) + \delta \right\|^2 - \frac{1}{2\beta} \|\nabla f(\theta)\|^2 .
\end{aligned}
$$

Next we notice that the contributions of the two terms $\frac{\beta}{2} \left\| \frac{1}{\beta} \nabla f(\theta) + \delta \right\|^2$ and $\frac{1}{2\beta} \|\nabla f(\theta)\|^2$ exactly match on the coordinates not touched by $\delta$. Hence everything outside the support of $\delta$ cancels out, which yields the desired conclusion. $\square$

We require another lemma which will be very useful in the analysis.

**Lemma 3.** *Let $\theta, g \in \mathbb{R}^N$ such that $\text{supp}(\theta) = S$, and let $S', S^*$ be some arbitrary subsets, with $|S'| = |S| > |S^*|$. Furthermore suppose that*

$$T_k(\theta + g) = (\theta + g)_{S'} .$$

*Then*

$$\left\| (\theta + g)_{S \setminus S'} \right\|^2 - \|g_{S \cup S'}\|^2 \le \left\| (\theta + g)_{Z \setminus S'} \right\|^2 - \|g_{S^*}\|^2 ,$$

*for some set $Z$ such that $|Z \setminus S'| \le 2|S^*|$.*

*Proof.* We prove this as follows. We write

$$\left\|(\theta+g)_{S\setminus S'}\right\|^2 - \|g_{S\cup S'}\|^2 = \left\|(\theta+g)_{(S^*\cup S)\setminus S'}\right\|^2 - \left\|(\theta+g)_{S^*\setminus(S\cup S')}\right\|^2 - \|g_{S\cup S'}\|^2$$

$$= \left\|(\theta+g)_{(S^*\cup S)\setminus S'}\right\|^2 - \|g_{S^*\setminus(S\cup S')}\|^2 - \|g_{S\cup S'}\|^2$$

$$= \left\|(\theta+g)_{(S^*\cup S)\setminus S'}\right\|^2 - \|g_{S^*\cup S\cup S'}\|^2$$

$$= \left\|(\theta+g)_{(S^*\cup S)\setminus S'}\right\|^2 - \|g_{S'\setminus(S^*\cup S)}\|^2 - \|g_{S^*\cup S}\|^2 \ .$$

Since

$$\|g_{S'\setminus(S^*\cup S)}\|^2 = \left\|(\theta+g)_{S'\setminus(S^*\cup S)}\right\|^2 \geq \left\|(\theta+g)_R\right\|^2 \ ,$$

where $R$ is a subset $R \subseteq (S^* \cup S) \setminus S'$ with $|R| = |S' \setminus (S^* \cup S)|$. Such a set definitely exists as

$$\left|(S^* \cup S) \setminus S'\right| \geq |S \setminus S'| = |S' \setminus S| \geq |S' \setminus (S^* \cup S)| = |R| \ .$$

Hence we obtain that

$$\left\|(\theta+g)_{S\setminus S'}\right\|^2 - \|g_{S\cup S'}\|^2 \leq \left\|(\theta+g)_{((S^*\cup S)\setminus S')\setminus R}\right\|^2 - \|g_{S^*\cup S}\|^2 \ .$$

Note that

$$\left|\left((S^* \cup S) \setminus S'\right) \setminus R\right| = \left|(S^* \cup S) \setminus S'\right| - |R| = \left|(S^* \cup S) \setminus S'\right| - |S' \setminus (S^* \cup S)|$$

$$\leq \left(|S^*| + |S \setminus S'|\right) - \left(|S' \setminus S| - |S^*|\right)$$

$$= 2|S^*| \ .$$

This concludes the proof. □

Using this we derive the following useful corollary.

**Corollary 2.** *Let $\theta, g \in \mathbb{R}^N$ such that $\operatorname{supp}(\theta) = S$, and let $S', S^*$ be arbitrary subsets, with $|S'| = |S| > |S^*|$. Furthermore suppose that*

$$T_k(\theta+g) = (\theta+g)_{S'} \ .$$

*Then one has that*

$$\left\|(\theta+g)_{S\setminus S'}\right\|^2 - \|g_{S\cup S'}\|^2 \leq \frac{2|S^*| + |\operatorname{supp}(\theta^*)|}{|S'| - |\operatorname{supp}(\theta^*)|} \cdot \left\|(\theta+g)_T - \theta^*\right\|^2 - \|g_{S^*}\|^2 \ ,$$

*for some $T$, such that $|T| \leq 2|S^*| + |\operatorname{supp}(\theta^*)| + |S'|$ and $\operatorname{supp}(\theta^*) \subseteq T$.*

*Proof.* Using Lemma 3 we can write

$$\left\|(\theta+g)_{S\setminus S'}\right\|^2 - \|g_{S\cup S'}\|^2 \leq \left\|(\theta+g)_{Z\setminus S'}\right\|^2 - \|g_{S^*}\|^2 \leq \left\|(\theta+g)_{(Z\cup\operatorname{supp}(\theta^*))\setminus S'}\right\|^2 - \|g_{S^*}\|^2$$

$$= \left\|(\theta+g)_{Z\cup\operatorname{supp}(\theta^*)\cup S'} - (\theta+g)_{S'}\right\|^2 - \|g_{S^*}\|^2 \ .$$

where $|Z \cup \operatorname{supp}(\theta^*) \cup S'| \leq 2|S^*| + |\operatorname{supp}(\theta^*)| + |S'|$. Applying Lemma 1 we furthermore obtain that

$$\left\|(\theta+g)_{Z\cup S^*\cup S'} - (\theta+g)_{S'}\right\|^2 \leq \frac{|Z \cup \operatorname{supp}(\theta^*) \cup S'| - |S'|}{|Z \cup \operatorname{supp}(\theta^*) \cup S'| - |\operatorname{supp}(\theta^*)|} \cdot \left\|(\theta+g)_{Z\cup\operatorname{supp}(\theta^*)\cup S'} - \theta^*\right\|^2$$

$$\leq \frac{2|S^*| + |\operatorname{supp}(\theta^*)|}{|S'| - |\operatorname{supp}(\theta^*)|} \cdot \left\|(\theta+g)_{Z\cup\operatorname{supp}(\theta^*)\cup S'} - \theta^*\right\|^2 \ .$$

□

We can now proceed with the main proof.

*Proof of Theorem 1.* For simplicity we first provide the proof for the deterministic version, which roughly follows the ideas described in [31]. Afterwards, we extend it to the stochastic setting. To simplify notation, throughout this proof we will use $\eta = \frac{1}{\beta}$.

Using Lemma 2 we can write that for the update

$$\delta = T_k(\theta - \eta\nabla f(\theta)) - \theta \ ,$$

we have

$$f\left(\theta'\right) \leq f\left(\theta\right) + \frac{1}{2\eta}\left\|\left(\frac{1}{\beta}\nabla f\left(\theta\right) + T_k\left(\theta - \eta\nabla f\left(\theta\right)\right) - \theta\right)_{\text{supp}(\delta)}\right\|^2 - \frac{1}{2\eta}\left\|\eta\nabla f\left(\theta\right)_{\text{supp}(\delta)}\right\|^2$$

$$= f\left(\theta\right) + \frac{1}{2\eta}\left\|\left(T_k\left(\theta - \eta\nabla f\left(\theta\right)\right) - \left(\theta - \eta\nabla f\left(\theta\right)\right)\right)_{\text{supp}(\delta)}\right\|^2 - \frac{1}{2\eta}\left\|\eta\nabla f\left(\theta\right)_{\text{supp}(\delta)}\right\|^2 .$$

At this point we use the fact that by definition $\text{supp}\left(\delta\right) = S' \cup S$. Furthermore we see that

$$\left\|\left(T_k\left(\theta - \eta\nabla f\left(\theta\right)\right) - \left(\theta - \eta\nabla f\left(\theta\right)\right)\right)_{S'\cup S}\right\|^2 = \left\|\left(\theta - \eta\nabla f\left(\theta\right)\right)_{S\setminus S'}\right\|^2$$

since $T_k\left(\theta - \eta\nabla f\left(\theta\right)\right)$ exactly matches $\theta - \eta\nabla f\left(\theta\right)$ for the coordinates in $S'$, and is 0 for all the others. Thus we have that

$$f\left(\theta'\right) \leq f\left(\theta\right) + \frac{1}{2\eta}\left(\left\|\left(\theta - \eta\nabla f\left(\theta\right)\right)_{S\setminus S'}\right\|^2 - \left\|\eta\nabla f\left(\theta\right)_{S\cup S'}\right\|^2\right) .$$

In order to apply the CPL property, we need to relate this to the contribution to the gradient norm given by the heavy signal $\|T_{k^*}\left(\nabla f\left(\theta\right)\right)\|^2$. Let $S^*$ be the support of $T_{k^*}\left(\nabla f\left(\theta\right)\right)$. We apply Corollary 2 to further bound

$$f\left(\theta'\right) \leq f\left(\theta\right) + \frac{1}{2\eta}\left(\frac{3k^*}{k - k^*}\left\|\left(\theta - \eta\nabla f\left(\theta\right)\right)_T - \theta^*\right\|^2 - \left\|\eta\nabla f\left(\theta\right)_{S^*}\right\|^2\right) ,$$

where $|T| \leq 3k^* + k$. Now we apply the CPL property as follows. From Lemma 6 we upper bound

$$\left\|\left(\theta - \eta\nabla f\left(\theta\right)\right)_T - \theta^*\right\|^2 \leq \left\|\left(\theta - \eta\nabla f\left(\theta\right)\right)_{T\cup S} - \theta^*\right\|^2$$

$$\leq \frac{8}{\alpha}\left(f\left(\theta - \eta\nabla f\left(\theta\right)_{T\cup S}\right) - f\left(\theta^*\right)\right)$$

$$\leq \frac{8}{\alpha}\left(f\left(\theta\right) - f\left(\theta^*\right)\right) ,$$

In the first inequality we used the fact that $\text{supp}\left(\theta^*\right) \subseteq T$. In the second one we applied Lemma 6. In the third one we applied $(3k^* + 2k, \beta)$-smoothness (since $|T \cup S| \leq 3k^* + 2k$) together with the fact that $\eta = 1/\beta$, and so $f\left(\theta - \eta\nabla f\left(\theta\right)_{T\cup S}\right) \leq f\left(\theta\right)$.

Similarly we apply the CPL inequality to conclude that

$$f\left(\theta'\right) \leq f\left(\theta\right) + \frac{1}{2\eta}\left(\frac{3k^*}{k - k^*}\cdot\frac{8}{\alpha}\left(f\left(\theta\right) - f\left(\theta^*\right)\right) - \eta^2\cdot\frac{\alpha}{2}\left(f\left(\theta\right) - f\left(\theta^*\right)\right)\right) .$$

Thus equivalently:

$$f\left(\theta'\right) - f\left(\theta^*\right) \leq \left(f\left(\theta\right) - f\left(\theta^*\right)\right)\left(1 + \frac{12k^*}{k - k^*}\cdot\frac{1}{\eta\alpha} - \frac{\eta\alpha}{4}\right)$$

$$= \left(f\left(\theta\right) - f\left(\theta^*\right)\right)\left(1 + \frac{12k^*}{k - k^*}\cdot\kappa - \frac{1}{4\kappa}\right) ,$$

where $\kappa = \beta/\alpha = 1/(\eta\alpha)$. Since $k \geq k^*\cdot\left(96\kappa^2 + 1\right)$ we equivalently have that $\frac{k^*}{k - k^*} \leq \frac{1}{96k^2}$ and thus $\frac{12k^*}{k - k^*}\cdot\kappa \leq \frac{1}{8\kappa}$. Hence

$$f\left(\theta'\right) - f\left(\theta^*\right) \leq \left(f\left(\theta\right) - f\left(\theta^*\right)\right)\cdot\left(1 - \frac{1}{8\kappa}\right) .$$

This shows that in $O\left(\kappa\ln\frac{f(\theta_0) - f(\theta^*)}{\epsilon}\right)$ we reach a point $\theta$ such that $f\left(\theta\right) - f\left(\theta^*\right) \leq \epsilon$, which concludes the proof.

We extend this proof to the stochastic version in Section A.5. $\qquad\square$

Now let us show how Corollary 1 follows from the same analysis.

*Proof of Corollary 1.* The proof extends from that to Theorem 1. The difference we need to handle is the error introduced by performing $\Delta_c$ passes through the data instead of a single stochastic gradient step. Suppose that before starting the dense training phase, the current iterate is $\theta$. The key is to bound the error introduced by performing these $\Delta_c$ passes instead of simply changing the iterate by $-\eta\nabla f(\theta)$, prior to applying the pruning step. To do so we first upper bound the change in the iterate after $\Delta_c$ passes, which lead to a new iterate $\widetilde{\theta}$. Since we perform $\Delta_c\cdot B$ iterations instead of a single one, we damp down our step size by setting $\eta' = \eta/(\Delta_c B)$,

and measure the movement we made compared to the one we would have made with a single deterministic gradient step.

Using the Lipschitz property of the functions in the decomposition, we see that each step changes our iterate by at most $\eta'L$ in $\ell_2$ norm. Hence over $\Delta_c$ passes through the data, each of which involves $B$ mini-batches, the total change in the iterate is at most:

$$\|\widetilde{\theta} - \theta\| \leq \Delta_c \cdot B \cdot \eta'L = \eta L .$$

Using the smoothness property of each $f_i$ this guarantees that for all the seen iterates $\widehat{\theta}$, the gradients of $f_i$'s never deviate significantly from their values at the original point $\theta$:

$$\|\nabla f_i(\widehat{\theta}) - \nabla f_i(\theta)\| \leq \beta\eta L ,$$

Thus if we interpret the scaled total movement in iterate $\frac{1}{\eta}(\theta - \widetilde{\theta})$ as a gradient mapping, it satisfies

$$\left\| \frac{1}{\eta}(\theta - \widetilde{\theta}) - \nabla f(\theta) \right\| \leq \beta\eta L .$$

Applying the analysis for the stochastic version of Theorem 1 we can treat the error in the gradient mapping exactly as the stochastic noise. For the specific choice of step size used there $\eta = 1/\Theta(\beta)$, we thus get that

$$\left\| \frac{1}{\eta}(\theta - \widetilde{\theta}) - \nabla f(\theta) \right\|^2 = O(L^2) ,$$

which enables us to conclude the analysis. Note that the steps performed during the sparse training phases do not affect convergence as they can only improve error in function value, which is the main quantity that our analysis tracks.

$\square$

## A.3 Stochastic IHT for Functions with Restricted Smoothness and Strong Convexity

Here we prove Theorem 2. Before proceeding with the proof, we provide a few useful statements related to the fact that $f$ is well conditioned along sparse directions.

**Lemma 4.** *If $f$ is $(2k + k^*, \beta)$-smooth and $(k + k^*, \alpha)$-strongly convex then*

$$f(\theta^*) \geq f(\theta) + \langle \nabla f(\theta)_{S \cup S' \cup S^*}, \theta^* - \theta \rangle + \frac{\alpha}{2} \|\theta - \theta^*\|^2 ,$$

$$f\left( \theta - \frac{1}{\beta}\nabla f(\theta)_{S \cup S' \cup S^*} \right) \leq f(\theta) - \frac{1}{2\beta} \|\nabla f(\theta)_{S \cup S' \cup S^*}\|^2 .$$

*Proof.* The former follows directly from the definition. For the latter we have

$$f\left( \theta - \frac{1}{\beta}\nabla f(\theta)_{S \cup S' \cup S^*} \right) \leq f(\theta) - \frac{1}{\beta} \langle \nabla f(\theta), \nabla f(\theta)_{S \cup S' \cup S^*} \rangle + \frac{\beta}{2} \|\nabla f(\theta)_{S \cup S' \cup S^*}\|^2$$

$$= f(\theta) - \frac{1}{2\beta} \|\nabla f(\theta)_{S \cup S' \cup S^*}\|^2 .$$

$\square$

Finally we can prove the main statement:

*Proof.* We will track progress by measuring the distance $\|\theta - \theta^*\|^2$. To do so we write

$$\|\theta' - \theta^*\|^2 = \|T_k(\theta - \eta g_\theta) - \theta^*\|^2$$
$$= \|(\theta - \eta g_\theta - \theta^*) + (T_k(\theta - \eta g_\theta) - (\theta - \eta g_\theta))\|^2$$
$$= \left\|(\theta - \eta g_\theta - \theta^*)_{S' \cup S^*} + \left(T_k((\theta - \eta g_\theta)_{S' \cup S^*}) - (\theta - \eta g_\theta)_{S' \cup S^*}\right)\right\|^2 .$$

The last identity follows from the fact that the term inside the norm is only supported at $S' \cup S^*$. Thus, after applying the triangle inequality, we obtain that:

$$\|\theta' - \theta^*\|^2 \leq \left( \|(\theta - \eta g_\theta)_{S' \cup S^*} - \theta^*\| + \left\|T_k((\theta - \eta g_\theta)_{S' \cup S^*}) - (\theta - \eta g_\theta)_{S' \cup S^*}\right\| \right)^2 .$$

The second term can now be bounded using Lemma 1. Since the sparsity of the projected point is $k + k^*$ we have that

$$\left\|T_k((\theta - \eta g_\theta)_{S' \cup S^*}) - (\theta - \eta g_\theta)_{S' \cup S^*}\right\|^2 \leq \frac{k + k^* - k}{k + k^* - k^*} \left\|(\theta - \eta g_\theta)_{S' \cup S^*} - \theta^*\right\|^2$$

$$= \frac{k^*}{k} \cdot \left\|(\theta - \eta g_\theta)_{S' \cup S^*} - \theta^*\right\|^2 .$$

Therefore:

$$\left\|\theta' - \theta^*\right\|^2 \leq \left(1 + \sqrt{\frac{s^*}{s}}\right)^2 \cdot \left\|(\theta - \eta g_\theta)_{S'\cup S^*} - \theta^*\right\|^2 \leq \left(1 + \sqrt{\frac{s^*}{s}}\right)^2 \cdot \left\|(\theta - \eta g_\theta)_{S\cup S'\cup S^*} - \theta^*\right\|^2 .$$

The second inequality follows from the fact that as we increase the support of $\theta - \eta g_\theta$ to also include $S\backslash(S'\cup S^*)$, the norm inside can only increase. Finally by expanding, we write:

$$\left\|(\theta - \eta g_\theta)_{S\cup S'\cup S^*} - \theta^*\right\|^2 = \left\|\theta - \eta\,(g_\theta)_{S\cup S'\cup S^*} - \theta^*\right\|^2$$

$$= \left\|\theta - \theta^*\right\|^2 - 2\eta\left\langle (g_\theta)_{S\cup S'\cup S^*}, \theta - \theta^*\right\rangle + \eta^2\left\|(g_\theta)_{S\cup S'\cup S^*}\right\|^2$$

$$\leq \left\|\theta - \theta^*\right\|^2 - 2\eta\left\langle \nabla f\,(\theta)_{S\cup S'\cup S^*}, \theta - \theta^*\right\rangle + 2\eta^2\left\|\nabla f\,(\theta)_{S\cup S'\cup S^*}\right\|^2$$

$$+ \underbrace{2\eta\left\langle (\nabla f\,(\theta) - g_\theta)_{S\cup S'\cup S^*}, \theta - \theta^*\right\rangle + 2\eta^2\left\|(\nabla f\,(\theta) - g_\theta)_{S\cup S'\cup S^*}\right\|^2}_{\zeta},$$

where we use $\zeta$ to denote the error term introduced by using stochastic gradients. Next we bound the fist part of the term above. To do so we use lemma to write

$$\left\|\theta - \theta^*\right\|^2 - 2\eta\left\langle \nabla f\,(\theta)_{S\cup S'\cup S^*}, \theta - \theta^*\right\rangle + 2\eta^2\left\|\nabla f\,(\theta)_{S\cup S'\cup S^*}\right\|^2$$

$$\leq \left\|\theta - \theta^*\right\|^2 - 2\eta\left(f\,(\theta) - f\,(\theta^*) - \frac{\alpha}{2}\left\|\theta - \theta^*\right\|^2\right) + 2\eta^2\left\|\nabla f\,(\theta)_{S\cup S'\cup S^*}\right\|^2$$

$$= \left\|\theta - \theta^*\right\|^2(1 - \eta\alpha) - 2\eta\left(f\,(\theta) - f\,(\theta^*) - \eta\left\|\nabla f\,(\theta)_{S\cup S'\cup S^*}\right\|^2\right)$$

$$= \left\|\theta - \theta^*\right\|^2(1 - \eta\alpha) - 2\eta\left(f\,(\theta) - f\,(\theta^*) - \frac{1}{2\beta}\left\|\nabla f\,(\theta)_{S\cup S'\cup S^*}\right\|^2\right) - 2\eta\cdot\left(\frac{1}{2\beta} - \eta\right)\left\|\nabla f\,(\theta)_{S\cup S'\cup S^*}\right\|^2$$

$$\leq \left\|\theta - \theta^*\right\|^2(1 - \eta\alpha) - 2\eta\cdot\left(\frac{1}{2\beta} - \eta\right)\left\|\nabla f\,(\theta)_{S\cup S'\cup S^*}\right\|^2 .$$

For the first inequality we used the restricted strong convexity property, while for the second we used the restricted smoothness property. Now for the error term we have that:

$$\mathbb{E}\left[\zeta|\theta\right] \leq 2\eta^2\sigma^2 ,$$

which enables us to conclude that

$$\mathbb{E}\left[\left\|\theta' - \theta^*\right\|^2\,\Big|\,\theta\right] \leq \left(1 + \sqrt{\frac{k^*}{k}}\right)^2 \cdot \left(\left\|\theta - \theta^*\right\|^2(1 - \eta\alpha) - 2\eta\cdot\left(\frac{1}{2\beta} - \eta\right)\left\|\nabla f\,(\theta)_{S\cup S'\cup S^*}\right\|^2 + 2\eta^2\sigma^2\right) .$$

Setting $\eta = \frac{1}{2\beta}$ this gives us that

$$\mathbb{E}\left[\left\|\theta' - \theta^*\right\|^2\,\Big|\,\theta\right] \leq \left(1 + \sqrt{\frac{s^*}{s}}\right)^2 \cdot \left(\left\|\theta - \theta^*\right\|^2\left(1 - \frac{1}{2}\cdot\frac{\alpha}{\beta}\right) + \frac{1}{2}\left(\frac{\sigma}{\beta}\right)^2\right) .$$

Thus for as long as $\left\|\theta - \theta^*\right\|^2 \geq 2\frac{\sigma^2}{\alpha\beta}$ we have that

$$\mathbb{E}\left[\left\|\theta' - \theta^*\right\|^2\,\Big|\,\theta\right] \leq \left(1 + \sqrt{\frac{k^*}{k}}\right)^2 \cdot \left(1 - \frac{1}{4}\cdot\frac{\alpha}{\beta}\right) \cdot \left\|\theta - \theta^*\right\|^2 .$$

We see that the expected squared distance contracts by setting the ratio $k/k^*$ sufficiently large. Indeed if $k \geq 81\,(\beta/\alpha)^2 \cdot k^*$ we have that:

$$\mathbb{E}\left[\left\|\theta' - \theta^*\right\|^2\,\Big|\,\theta\right] \leq \left(1 + \frac{1}{9}\cdot\frac{\alpha}{\beta}\right)^2 \cdot \left(1 - \frac{1}{4}\cdot\frac{\alpha}{\beta}\right) \cdot \left\|\theta - \theta^*\right\|^2 \leq \left(1 - \frac{1}{36}\cdot\frac{\alpha}{\beta}\right) \cdot \left\|\theta - \theta^*\right\|^2 .$$

Taking expectation over the entire history of iterates we can thus conclude that after $T = O\left(\frac{\beta}{\alpha}\cdot\ln\frac{\|\theta_0 - \theta^*\|^2}{\epsilon/\beta + \sigma^2/\alpha\beta}\right)$ iterations we obtain

$$\mathbb{E}\left[\left\|\theta_T - \theta^*\right\|^2\right] \leq \frac{\epsilon}{\beta} + \frac{2\sigma^2}{\alpha\beta} .$$

Applying the restricted smoothness property, this also gives us that:

$$\mathbb{E}\left[f\,(\theta_T) - f\,(\theta^*)\right] \leq \epsilon + \frac{2\sigma^2}{\alpha} .$$

which is what we wanted. □

## A.4 Finding a Sparse Nearly-Stationary Point

Here we prove Theorem 3. For simplicity, we first prove the deterministic version of the theorem, where $\sigma = 0$. We show how to extend this proof to the stochastic version in Section A.5.

*Proof.* The proof is similar to that of Theorem 1, but in addition requires that the algorithm alternates standard IHT steps with optimizing inside the support of the iterate. More precisely given a current iterate supported at $S$, we additionally run an inner loop which optimizes only over the coordinate in $S$, seeking a near stationary point $\theta$ such that $\left\| \nabla f\left(\theta\right)_S \right\| \leq \epsilon$. Thus in our analysis we can assume that before performing a gradient, followed by a pruning step, our current iterate supported at $S$ satisfies $\left\| \nabla f\left(\theta\right)_S \right\| \leq \epsilon$. Hence following the previous analyses and setting $\eta = 1/\beta$, we have:

$$
f\left(\theta'\right) \leq f\left(\theta\right) + \frac{1}{2\eta}\left(\left\|\left(\theta - \eta\nabla f\left(\theta\right)\right)_{S\setminus S'}\right\|^2 - \left\|\eta\nabla f\left(\theta\right)_{S\cup S'}\right\|^2\right)
$$

$$
\leq f\left(\theta\right) + \frac{1}{2\eta}\left(2\left\|\theta_{S\setminus S'}\right\|^2 + 2\eta^2\epsilon^2 - \left\|\eta\nabla f\left(\theta\right)\right\|_\infty^2\right) .
$$

For the second inequality we first applied triangle inequality together with the near stationarity condition for $\theta$ to bound

$$
\left\|\left(\theta - \eta\nabla f\left(\theta\right)\right)_{S\setminus S'}\right\| \leq \left\|\theta_{S\setminus S'}\right\| + \left\|\eta\nabla f\left(\theta\right)_{S\setminus S'}\right\| \leq \left\|\theta_{S\setminus S'}\right\| + \eta\epsilon ,
$$

then applied $\left(a + b\right)^2 \leq 2a^2 + 2b^2$. In addition we used the fact that

$$
\left\|\nabla f\left(\theta\right)_{S\cup S'}\right\|_\infty = \left\|\nabla f\left(\theta\right)\right\|_\infty .
$$

This follows from a simple case analysis. If one of the coordinates of the gradient with the largest absolute value lies in $S \cup S'$, we are done. Otherwise, we have two possibilities. Either $S'$ is different from $S$, so $S \cup S'$ contains one coordinate outside of $S$. Since these coordinates are obtained by hard thresholding and $\theta$ is supported only at $S$, the absolute value of $\nabla f\left(\theta\right)$ at the largest coordinate in $S' \setminus S$ must be at least as large as the largest in $S^* \setminus S$, which yields our claim. Otherwise we have that $S = S'$, which means that the pruning step did not change the support, and thus

$$
\left\|\eta\nabla f\left(\theta\right)_{\overline{S}}\right\|_\infty \leq \min_{i \in S}\left|\theta - \eta\nabla f\left(\theta\right)\right|_i \leq R_\infty + \eta\epsilon
$$

which guarantees that

$$
\left\|\nabla f\left(\theta\right)\right\|_\infty = \max\left\{\left\|\nabla f\left(\theta\right)_S\right\|_\infty, \left\|\nabla f\left(\theta\right)_{\overline{S}}\right\|_\infty\right\}
$$

$$
\leq \max\left\{\epsilon, \epsilon + \frac{R_\infty}{\eta}\right\} = \epsilon + \beta R_\infty ,
$$

and so we are done.

In the former case we thus see that

$$
\frac{\eta}{2}\left\|\nabla f\left(\theta\right)\right\|_\infty^2 \leq f\left(\theta\right) - f\left(\theta'\right) + \left(\frac{\left\|\theta_{S\setminus S'}\right\|^2}{\eta} + \eta\epsilon^2\right) \leq f\left(\theta\right) - f\left(\theta'\right) + \left(\frac{kR_\infty^2}{\eta} + \eta\epsilon^2\right) .
$$

Telescoping over $T$ iterations we see that

$$
\frac{\eta}{2}\sum_{t=0}^{T-1}\left\|\nabla f\left(\theta_t\right)\right\|_\infty^2 \leq f\left(\theta_0\right) - f\left(\theta_T\right) + T \cdot \left(\frac{kR_\infty^2}{\eta} + \eta\epsilon^2\right)
$$

and so returning a random point $\theta$ among those witnessed during the algorithm we have

$$
\mathbb{E}\left[\left\|\nabla f\left(\theta\right)\right\|_\infty^2\right] \leq \frac{2\left(f\left(\theta_0\right) - f\left(\theta_T\right)\right)}{\eta T} + 2\left(\frac{kR_\infty^2}{\eta^2} + \epsilon^2\right)
$$

By AM-QM,

$$
\mathbb{E}\left[\left\|\nabla f\left(\theta\right)\right\|_\infty\right] \leq \sqrt{\mathbb{E}\left[\left\|\nabla f\left(\theta\right)\right\|_\infty^2\right]} ,
$$

which enables us to conclude that after sufficiently many iterations we are guaranteed to find a point such that

$$
\left\|\nabla f\left(\theta\right)\right\|_\infty = O\left(\frac{R_\infty\sqrt{k}}{\eta} + \epsilon\right) = O\left(\beta R_\infty\sqrt{k} + \epsilon\right) .
$$

$\square$

## A.5 Deferred Proofs

**Proof of Lemma 1.** Our proofs crucially rely on the following lemma. Intuitively it shows that projecting a vector $\theta$ onto the non-convex set of sparse vectors does not increase the distance to the optimum by too much. While this is indeed always true for projections onto convex sets, in this case we can provably show that the possible increase in distance is small.

*Proof.* We have that the function

$$h(k) = \frac{\|T_k(\theta) - \theta\|^2}{n-k}$$

is non-increasing. Indeed, using more nonzeros can only decrease the ratio. Thus

$$\frac{\|T_k(\theta) - \theta\|^2}{n-k} \leq \frac{\|T_{k^*}(\theta) - \theta\|^2}{n-k^*} \leq \frac{\|\theta^* - \theta\|^2}{n-k^*} \, ,$$

where the last inequality follows from the fact that among all $k$-sparse vectors, $T_k(\theta)$ minimizes the distance to $\theta$. □

**Proof of Theorem 1 (stochastic version).** Next we extend the proof of Theorem 1 to the case when only stochastic gradients are available.

*Proof.* Similarly to before we can write

$$f(\theta') \leq f(\theta) + \frac{\beta}{2}\left\| \frac{1}{\beta}\nabla f(\theta) + (T_k(\theta - \eta g_\theta) - \theta) \right\|^2 - \frac{1}{2\beta}\|\nabla f(\theta)\|^2$$

$$= f(\theta) + \frac{\beta}{2}\left\| \left(T_k(\theta - \eta g_\theta) - \left(\theta - \frac{1}{\beta}\nabla f(\theta)\right)\right)_{S\cup S'} \right\|^2 - \frac{1}{2\beta}\|\nabla f(\theta)_{S\cup S'}\|^2$$

$$\leq f(\theta) + \frac{\beta}{2}\left( \|(T_k(\theta - \eta g_\theta) - (\theta - \eta g_\theta))_{S\cup S'}\| + \left\| \left(\eta g_\theta - \frac{1}{\beta}\nabla f(\theta)\right)_{S\cup S'} \right\| \right)^2 - \frac{1}{2\beta}\|\nabla f(\theta)_{S\cup S'}\|^2$$

$$\leq f(\theta) + \beta\|(T_k(\theta - \eta g_\theta) - (\theta - \eta g_\theta))_{S\cup S'}\|^2 + \beta\left\| \left(\eta g_\theta - \frac{1}{\beta}\nabla f(\theta)\right)_{S\cup S'} \right\|^2 - \frac{1}{2\beta}\|\nabla f(\theta)_{S\cup S'}\|^2$$

$$= f(\theta) + \beta\left\| (\theta - \eta g_\theta)_{S\setminus S'} \right\|^2 + \beta\left\| \left(\eta g_\theta - \frac{1}{\beta}\nabla f(\theta)\right)_{S\cup S'} \right\|^2 - \frac{1}{2\beta}\|\nabla f(\theta)_{S\cup S'}\|^2 \, .$$

Next we apply Corollary 2 to further bound

$$\left\| (\theta - \eta g_\theta)_{S\setminus S'} \right\|^2 \leq \frac{2|S^*| + |\mathrm{supp}(\theta^*)|}{|S'| - |\mathrm{supp}(\theta^*)|} \cdot \left\| (\theta - \eta g_\theta)_T - \theta^* \right\|^2 - \left\| \eta(g_\theta)_{S*} \right\|^2 + \left\| \eta(g_\theta)_{S\cup S'} \right\|^2$$

$$= \frac{3k^*}{k-k^*} \cdot \left\| (\theta - \eta g_\theta)_T - \theta^* \right\|^2 - \left\| \eta(g_\theta)_{S*} \right\|^2 + \left\| \eta(g_\theta)_{S\cup S'} \right\|^2 \, ,$$

where $|T| \leq 3k^* + k$. Thus

$$f(\theta') \leq f(\theta) + \beta\left( \frac{3k^*}{k-k^*} \cdot \left\| (\theta - \eta g_\theta)_T - \theta^* \right\|^2 - \left\| \eta(g_\theta)_{S*} \right\|^2 + \left\| \eta(g_\theta)_{S\cup S'} \right\|^2 \right)$$

$$+ \beta\left\| \left(\eta g_\theta - \frac{1}{\beta}\nabla f(\theta)\right)_{S\cup S'} \right\|^2 - \frac{1}{2\beta}\|\nabla f(\theta)_{S\cup S'}\|^2 \, .$$

Taking expectations, and applying Lemma 5 we see that

$$\mathbb{E}\left[ f(\theta') - f^*|\theta \right] \leq \mathbb{E}\left[ f(\theta) - f^*|\theta \right]$$

$$+ \beta\left( \frac{3k^*}{k-k^*} \cdot \left\| (\theta - \eta\nabla f(\theta))_T - \theta^* \right\|^2 - \left\| \eta\nabla f(\theta)_{S*} \right\|^2 + \left\| \eta\nabla f(\theta)_{S\cup S'} \right\|^2 \right)$$

$$+ \beta\left\| \left(\eta\nabla f(\theta) - \frac{1}{\beta}\nabla f(\theta)\right)_{S\cup S'} \right\|^2 - \frac{1}{2\beta}\|\nabla f(\theta)_{S\cup S'}\|^2 + 2\beta\eta^2\sigma^2$$

$$= \mathbb{E}\left[ f(\theta) - f^*|\theta \right] + \beta\left( \frac{3k^*}{k-k^*} \cdot \left\| (\theta - \eta\nabla f(\theta))_T - \theta^* \right\|^2 - \left\| \eta\nabla f(\theta)_{S*} \right\|^2 \right)$$

$$+ \|\nabla f(\theta)_{S\cup S'}\|^2\left( \beta\eta^2 + \beta\left(\eta - \frac{1}{\beta}\right)^2 - \frac{1}{2\beta} \right) + 2\beta\eta^2\sigma^2$$

$$\leq \mathbb{E}\left[ f(\theta) - f^*|\theta \right] + \beta\left( \frac{3k^*}{k-k^*} \cdot \left\| (\theta - \eta\nabla f(\theta))_T - \theta^* \right\|^2 - \left\| \eta\nabla f(\theta)_{S*} \right\|^2 \right) + 2\beta\eta^2\sigma^2 \, ,$$

where the last inequality follows from setting $\eta = \frac{1}{2\beta}$, which makes $\beta\eta^2 + \beta\left(\eta - \frac{1}{\beta}\right)^2 - \frac{1}{2\beta} = 0$.

Finally, repeating the argument used for the deterministic proof in Section A.2, we further bound

$$\mathbb{E}\left[f\left(\theta'\right) - f^*|\theta\right] \leq f\left(\theta\right) - f^* + \beta\left(\frac{3k^*}{k - k^*} \cdot \frac{8}{\alpha}\left(f\left(\theta\right) - f\left(\theta^*\right)\right) - \eta^2 \cdot \frac{\alpha}{2}\left(f\left(\theta\right) - f\left(\theta^*\right)\right)\right) + 2\beta\eta^2\sigma^2$$

$$= \left(f\left(\theta\right) - f^*\right)\left(1 + \frac{24k^*}{k - k^*} \cdot \frac{\beta}{\alpha} - \frac{\beta\eta^2\alpha}{2}\right) + 2\beta\eta^2\sigma^2$$

$$= \left(f\left(\theta\right) - f^*\right)\left(1 + \frac{24k^*}{k - k^*} \cdot \frac{\beta}{\alpha} - \frac{\alpha}{8\beta}\right) + \frac{\sigma^2}{2\beta} .$$

Setting $k = k^* \cdot \left(384\left(\frac{\beta}{\alpha}\right)^2 + 1\right)$ we have $\frac{24k^*}{k - k^*} \cdot \frac{\beta}{\alpha} \leq 24 \cdot \frac{1}{384 \cdot (\beta/\alpha)^2} \cdot \frac{\beta}{\alpha} = \frac{1}{16} \cdot \frac{\alpha}{\beta}$, so

$$\mathbb{E}\left[f\left(\theta'\right) - f^*|\theta\right] \leq \left(f\left(\theta\right) - f^*\right)\left(1 - \frac{\alpha}{16\beta}\right) + \frac{\sigma^2}{2\beta} .$$

Thus for as long as

$$\frac{\sigma^2}{2\beta} \leq \left(f\left(\theta\right) - f^*\right) \cdot \frac{\alpha}{32\beta} \iff f\left(\theta\right) - f^* \geq 16 \cdot \frac{\sigma^2}{\alpha}$$

one has that

$$\mathbb{E}\left[f\left(\theta'\right) - f^*|\theta\right] \leq \left(f\left(\theta\right) - f^*\right)\left(1 - \frac{\alpha}{32\beta}\right) .$$

Taking expectation over the entire history, this shows that after $T = O\left(\left(\frac{\beta}{\alpha}\right)\ln\frac{f(\theta_0) - f^*}{\epsilon}\right)$ iterations we obtain an iterate $\theta_T$ such that

$$\mathbb{E}\left[f\left(\theta_T\right) - f^*\right] \leq \epsilon + \frac{16\sigma^2}{\alpha} ,$$

which concludes the proof. $\qquad\square$

**Proof of Theorem 3 (stochastic version).** We also provide the proof for the stochastic version of Theorem 3.

*Proof.* We follow the steps of the proof provided in Section A.4. More precisely we write:

$$f\left(\theta'\right) \leq f\left(\theta\right) + \frac{\beta}{2}\left\|\frac{1}{\beta}\nabla f\left(\theta\right) + \left(T_k\left(\theta - \eta g_\theta\right) - \theta\right)\right\|^2 - \frac{1}{2\beta}\left\|\nabla f\left(\theta\right)\right\|^2$$

$$= f\left(\theta\right) + \frac{\beta}{2}\left\|\left(T_k\left(\theta - \eta g_\theta\right) - \left(\theta - \frac{1}{\beta}\nabla f\left(\theta\right)\right)\right)_{S \cup S'}\right\|^2 - \frac{1}{2\beta}\left\|\nabla f\left(\theta\right)_{S \cup S'}\right\|^2$$

$$\leq f\left(\theta\right) + \frac{\beta}{2}\left(\left\|\left(T_k\left(\theta - \eta g_\theta\right) - \left(\theta - \eta g_\theta\right)\right)_{S \cup S'}\right\| + \left\|\left(\eta g_\theta - \frac{1}{\beta}\nabla f\left(\theta\right)\right)_{S \cup S'}\right\|\right)^2 - \frac{1}{2\beta}\left\|\nabla f\left(\theta\right)_{S \cup S'}\right\|^2$$

$$\leq f\left(\theta\right) + \beta\left\|\left(T_k\left(\theta - \eta g_\theta\right) - \left(\theta - \eta g_\theta\right)\right)_{S \cup S'}\right\|^2 + \beta\left\|\left(\eta g_\theta - \frac{1}{\beta}\nabla f\left(\theta\right)\right)_{S \cup S'}\right\|^2 - \frac{1}{2\beta}\left\|\nabla f\left(\theta\right)_{S \cup S'}\right\|^2$$

$$= f\left(\theta\right) + \beta\left\|\left(\theta - \eta g_\theta\right)_{S \setminus S'}\right\|^2 + \beta\left\|\left(\eta g_\theta - \frac{1}{\beta}\nabla f\left(\theta\right)\right)_{S \cup S'}\right\|^2 - \frac{1}{2\beta}\left\|\nabla f\left(\theta\right)_{S \cup S'}\right\|^2 ,$$

where we used the inequality $\left(a + b\right)^2 \leq 2a^2 + 2b^2$. Applying Lemma 5 and using the fact that $\left\|\nabla f\left(\theta\right)_S\right\| \leq \epsilon$, we see that setting $\eta = 1/\beta$ we obtain:

$$\mathbb{E}\left[f\left(\theta'\right)\right] \leq f\left(\theta\right) + \beta\left(\left\|\left(\theta - \eta\nabla f\left(\theta\right)\right)_{S \setminus S'}\right\|^2 + \sigma^2\right)$$

$$+ \beta\left(\left\|\left(\left(\eta - \frac{1}{\beta}\right)\nabla f\left(\theta\right)\right)_{S \cup S'}\right\|^2 + \sigma^2\right) - \frac{1}{2\beta}\left\|\nabla f\left(\theta\right)_{S \cup S'}\right\|^2$$

$$\leq f\left(\theta\right) + 2\beta\left(\left\|\theta_{S \setminus S'}\right\|^2 + \frac{\epsilon^2}{\beta^2} + \sigma^2\right) - \frac{1}{2\beta}\left\|\nabla f\left(\theta\right)_{S \cup S'}\right\|^2$$

$$\leq f\left(\theta\right) + 2\beta\left(kR_\infty^2 + \frac{\epsilon^2}{\beta^2} + \sigma^2\right) - \frac{1}{2\beta}\left\|\nabla f\left(\theta\right)\right\|_\infty^2 ,$$

where again we used the fact that $\left\|\nabla f\left(\theta\right)_{S\cup S'}\right\|_{\infty} = \left\|\nabla f\left(\theta\right)\right\|_{\infty}$, unless $\left\|\nabla f\left(\theta\right)\right\|_{\infty} \leq \frac{R_{\infty}}{\eta} = \beta R_{\infty}$. Thus, telescoping over $T$ iterations we see that

$$\frac{1}{2\beta} \sum_{t=0}^{T-1} \left\|\nabla f\left(\theta_t\right)\right\|_{\infty}^2 \leq f\left(\theta_0\right) - f\left(\theta_T\right) + T \cdot 2\beta \left(kR_{\infty}^2 + \beta^{-2}\epsilon^2 + \sigma^2\right)$$

and so returning a random point $\theta$ among those witnessed during the algorithm we have

$$\mathbb{E}\left[\left\|\nabla f\left(\theta\right)\right\|_{\infty}^2\right] \leq \frac{2\beta\left(f\left(\theta_0\right) - f\left(\theta_T\right)\right)}{T} + 2\left(\beta^2 kR_{\infty}^2 + \epsilon^2 + \beta^2\sigma^2\right)$$

By AM-QM,

$$\mathbb{E}\left[\left\|\nabla f\left(\theta\right)\right\|_{\infty}\right] \leq \sqrt{\mathbb{E}\left[\left\|\nabla f\left(\theta\right)\right\|_{\infty}^2\right]}$$

which enables us to conclude that after sufficiently many iterations we are guaranteed to find a point such that

$$\left\|\nabla f\left(\theta\right)\right\|_{\infty} = O\left(\beta R_{\infty}\sqrt{k} + \beta\sigma + \epsilon\right) .$$

$\square$

## Miscellaneous Proofs.

**Lemma 5.** *Let $\sigma > 0$ and let $g_\theta$ be a stochastic gradient satisfying standard conditions:*

$$\mathbb{E}[g_\theta|\theta] = \nabla f(\theta)],$$

*and*

$$\mathbb{E}[\|g_\theta - \nabla f(\theta)\|^2] \leq \sigma^2 .$$

*Then for any vector $a \in \mathbb{R}^N$ and any subset $S$ of coordinates:*

$$\left\|\left(\nabla f\left(\theta\right) + a\right)_S\right\|^2 \leq \mathbb{E}\left[\left\|\left(g_\theta + a\right)_S\right\|^2 \Big| \theta\right] \leq \left\|\left(\nabla f\left(\theta\right) + a\right)_S\right\|^2 + \sigma^2 .$$

*Proof.* We expand the norm under the expected value as:

$$\mathbb{E}\left[\left\|\left(g_\theta + a\right)_S\right\|^2 \Big| \theta\right] = \mathbb{E}\left[\left\|\left(\nabla f\left(\theta\right) + a\right)_S + \left(g_\theta - \nabla f\left(\theta\right)\right)_S\right\|^2 \Big| \theta\right]$$

$$= \mathbb{E}\left[\left\|\left(\nabla f\left(\theta\right) + a\right)_S\right\|^2 + \left\|\left(g_\theta - \nabla f\left(\theta\right)\right)_S\right\|^2 + 2\left\langle\left(\nabla f\left(\theta\right) + a\right)_S, \left(g_\theta - \nabla f\left(\theta\right)\right)_S\right\rangle \Big| \theta\right]$$

$$= \left\|\left(\nabla f\left(\theta\right) + a\right)_S\right\|^2 + \mathbb{E}\left[\left\|\left(g_\theta - \nabla f\left(\theta\right)\right)_S\right\|^2 \Big| \theta\right]$$

$$+ \mathbb{E}\left[2\left\langle\left(\nabla f\left(\theta\right) + a\right)_S, \left(g_\theta - \nabla f\left(\theta\right)\right)_S\right\rangle \Big| \theta\right]$$

$$\leq \left\|\left(\nabla f\left(\theta\right) + a\right)_S\right\|^2 + \sigma^2 .$$

From the the chain of equalities above, we can trivially lower bound the expectation by $\left\|\left(\nabla f\left(\theta\right) + a\right)_S\right\|^2$, which gives us what we needed. $\square$

A crucial step in the proof of Theorem 1 requires upper bounding the $\ell_2$ distance to the closest global optimizer by the difference in function value. In general, it is known that this is implied by the Polyak-Łojasiewicz condition, and so it automatically holds for the stronger concentrated Polyak-Łojasiewicz condition. We reproduce the proof from [34] for completeness.

**Lemma 6.** *(From Polyak-Łojasiewicz to quadratic growth) Let $f : \mathbb{R}^N \to \mathbb{R}$ be a function satisfying the Polyak-Łojasiewicz inequality*

$$\left\|\nabla f\left(\theta\right)\right\|^2 \geq \frac{\alpha}{2}\left(f\left(\theta\right) - f^*\right) .$$

*Then there exists a global minimizer $\theta^*$ of $f$ such that*

$$f\left(\theta\right) - f^* \geq \frac{\alpha}{8}\left\|\theta - \theta^*\right\|^2 .$$

*Proof.* Let $g\left(\theta\right) = \sqrt{f\left(\theta\right) - f^*}$ for which we have

$$\nabla g\left(\theta\right) = \frac{1}{2\sqrt{f\left(\theta\right) - f^*}}\nabla f\left(\theta\right) .$$

Using the PL condition we have

$$\|\nabla g\left(\theta\right)\|^2 = \frac{1}{4\left(f\left(\theta\right)-f^*\right)} \cdot \|\nabla f\left(\theta\right)\|^2 \geq \frac{1}{4\left(f\left(\theta\right)-f^*\right)} \cdot \frac{\alpha}{2} \cdot \left(f\left(\theta\right)-f^*\right) = \frac{\alpha}{8} \cdot$$

Now starting at some $\theta_0$, we consider the dynamic $\dot{\theta} = -\nabla g\left(\theta\right)$. We see that this always decreases function value until it reaches some $\theta_T$ for which $\nabla g\left(\theta_T\right) = 0$ and hence by the PL inequality, $\theta_T$ is a minimizer i.e. $f\left(\theta_T\right) = f^*$. Now we can write

$$g\left(\theta_T\right) = g\left(\theta_0\right) + \int_0^T \left\langle \nabla g\left(\theta_t\right), \dot{\theta}_t \right\rangle dt = g\left(\theta_0\right) + \int_0^T \left\langle \nabla g\left(\theta_t\right), -\nabla g\left(\theta_t\right) \right\rangle dt$$

$$= g\left(\theta_0\right) - \int_0^T \|\nabla g\left(\theta_t\right)\|^2 \, dt \; .$$

Thus

$$g\left(\theta_0\right) - g\left(\theta_T\right) = \int_0^T \|\nabla g\left(\theta_t\right)\|^2 \, dt \geq \sqrt{\frac{\alpha}{8}} \cdot \int_0^T \|\nabla g\left(\theta_t\right)\| \, dt = \sqrt{\frac{\alpha}{8}} \cdot \int_0^T \left\|\dot{\theta}_t\right\| dt \; ,$$

where we used our lower bound on the norm of $\nabla g\left(\theta\right)$. Finally, we use the fact that the last integral lower bounds the total movement of $\theta$ as it moves from $\theta_0$ to $\theta_T$. Thus

$$\int_0^T \left\|\dot{\theta}_t\right\| dt \geq \|\theta_0 - \theta_T\| \; ,$$

so

$$g\left(\theta_0\right) - g\left(\theta_T\right) \geq \sqrt{\frac{\alpha}{8}} \|\theta_0 - \theta_T\| \; ,$$

which enables us to conclude that

$$f\left(\theta_0\right) - f^* \geq \frac{\alpha}{8} \|\theta_0 - \theta_T\|^2 \; ,$$

where $\theta_T$ is some global minimizer of $f$. This concludes the proof. $\qquad\square$

## B   Additional Experiments

### B.1   CIFAR-100 Experiments

We tested the IHT pruning algorithm on the CIFAR-100 dataset [36]. We used the WideResNet-28-10 architecture [59], which to our knowledge gives state-of-the-art performance on this dataset. Models were trained for a fixed 200 epochs, using Stochastic Gradient Descent (SGD) with momentum, and a stepwise decreasing learning rate scheduler. For our main experiment, after a warm-up period of ten epochs, we alternated sparse and dense phases of 20 epochs until epoch 170, at which point we allowed the sparse model to train to convergence for 30 more epochs. In addition, we found a small benefit to resetting momentum to 0 at each transition from sparse to dense, and we have done so throughout the trials. These experiments were replicated starting from three different seeds, and we report average results and their standard deviations, and they are shown in table 5 in rows labeled "AC/DC-20".

We further explored the possibility of using even smaller dense phases to further reduce training FLOPs. These trials, presented in 5 show the results of reducing the sparse phase from 20 to 14 or even 7 epochs while keeping the length of dense phases the same, but increasing the number of total epochs to roughly match overall FLOPs (thus increasing the number of sparse and dense phases); overall we train the AC/DC-14 runs for 225 epochs and AC/DC-7 runs for 240 epochs (vs 200 for AC/DC-20). Our experiments show that it is possible to obtain competitive accuracy results with shorter dense phases, and, at higher sparsities, a further FLOP reduction - however, reducing the dense phase too much may lead to some accuracy degradation at higher sparsities. These results suggest that a further reduction in FLOPs is possible by adjusting the length of the dense phase and the overall epochs. We emphasize that $(i)$ these gains are only theoretical until hardware is available that can take advantage of sparsity in training and $(ii)$ these results, although promising, are highly preliminary; additionally, each trial was only run once due to timing constraints.

We compare our results with Gradual Magnitude Pruning [62]. To our knowledge, we are the first to release CIFAR-100 pruning results for this network architecture, and GMP was chosen as a baseline due to its generally strong performance against a range of other approaches [21]. We obtain the GMP baseline by training on WideResNet architecture at full density for 50 epochs, then gradually increase the sparsity over the next 100 before allowing the model to converge for the final 50, matching the 200 training epochs of AC/DC. We further validated this baseline by training dense WideResNet-28-10 models for 200 epochs and then gradually pruning over 50 and finetuning over 50 more, for a total of 300 epochs, which gave similar performance at the cost of greater FLOPs and training time.

The results are shown in Table 5. We see that AC/DC pruning significantly outperforms Gradual Magnitude Pruning at all sparsity levels tested, and further that AC/DC models pruned at 50% and 75% even outperform the dense baseline, while the models pruned at 90% at least match it.

Table 5: CIFAR-100 Sparsity results on WideResNet

| Method | Top-1 Acc. (%) | Sparsity | GFLOPs Inference | EFLOPs Train |
|--------|------------|----------|------------------|--------------|
| Dense | $79.0 \pm 0.25$ | 0% | 11.9 | 0.36 |
| AC/DC-20 | $79.6 \pm 0.17$ | 49.98% | $0.50\times$ | $0.72\times$ |
| AC/DC-14 | **79.99** | 49.98% | $0.50\times$ | $0.75\times$ |
| AC/DC-7 | 79.92 | 49.98% | $0.49\times$ | $0.73\times$ |
| GMP | $79.2 \pm 0.17$ | 49.98% | $0.46\times$ | $1.64\times$ |
| AC/DC-20 | $\mathbf{80.0 \pm 0.17}$ | 74.96% | $0.29\times$ | $0.6\times$ |
| AC/DC-14 | 79.5 | 74.96% | $0.29\times$ | $0.59\times$ |
| AC/DC-7 | 79.7 | 74.96% | $0.29\times$ | $0.54\times$ |
| GMP | $78.9 \pm 0.14$ | 74.96% | $0.26\times$ | $1.52\times$ |
| AC/DC-20 | $\mathbf{79.1 \pm 0.07}$ | 89.96% | $0.14\times$ | $0.51\times$ |
| AC/DC-14 | 79.0 | 89.96% | $0.14\times$ | $0.47\times$ |
| AC/DC-7 | 78.4 | 89.96% | $0.14\times$ | $0.39\times$ |
| GMP | $77.7 \pm 0.23$ | 89.96% | $0.08\times$ | $1.44\times$ |
| AC/DC-20 | $78.2 \pm 0.12$ | 94.95% | $0.08\times$ | $0.47\times$ |
| AC/DC-14 | **78.5** | 94.95% | $0.08\times$ | $0.41\times$ |
| AC/DC-7 | 77.8 | 94.95% | $0.08\times$ | $0.33\times$ |
| GMP | $76.6 \pm 0.07$ | 94.95% | $0.07\times$ | $1.41\times$ |

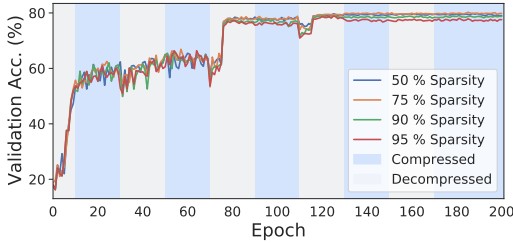

(a) Sparsity pattern and test accuracy

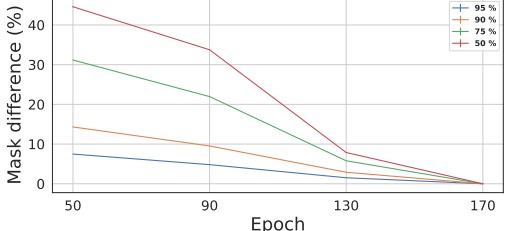

(b) Relative change in consecutive masks

Figure 4: Validation accuracy and sparsity during training, together with differences in consecutive masks for WideResNet on CIFAR-100 using AC/DC.

## B.2   ResNet50 on ImageNet

**Performance of dense models.** The AC/DC method has an advantage over other pruning methods of obtaining *both* sparse and dense models. The performance of the dense baseline can be recovered after fine-tuning the resulting AC/DC dense model, for a small number of epochs. Namely, we start from the best dense baseline, which is usually obtained after 85 epochs, and replace the final compression phase of 15 epochs with regular dense training; we use the same learning rate scheduler and keep all other training hyper-parameters the same. For 80% sparsity we recover the dense baseline accuracy completely, while for 90% we are slightly below the baseline by 0.3%. We note that for 90% sparsity, when the first and last layers are dense, our fine-tuned dense model recovers the baseline accuracy fully. The results for the dense models, together with the baseline accuracy, are presented in Table 6, where ($\star$) denotes that the first and last layers of the network are dense.

Table 6: AC/DC Dense ResNet50

| Target Sparsity | Accuracy Dense (%) | Accuracy Finetuned (%) |
|-----------------|--------------------|------------------------|
| 0% | 76.84 | - |
| 80% | $73.82 \pm 0.02$ | $76.83 \pm 0.07$ |
| 90% | $73.25 \pm 0.16$ | $76.56 \pm 0.1$ |
| 90%$^\star$ | 73.66 | 76.85 |

Table 7: AC/DC Dense MobileNetV1

| Sparsity | Accuracy Dense (%) | Accuracy Finetuned (%) |
|----------|--------------------|------------------------|
| 0% | 71.78 | - |
| 75% | $68.55 \pm 0.2$ | $71.63 \pm 0.1$ |
| 90% | $67.47 \pm 0.13$ | $70.86 \pm 0.08$ |
| 90%$^\star$ | 67.65 | 70.97 |

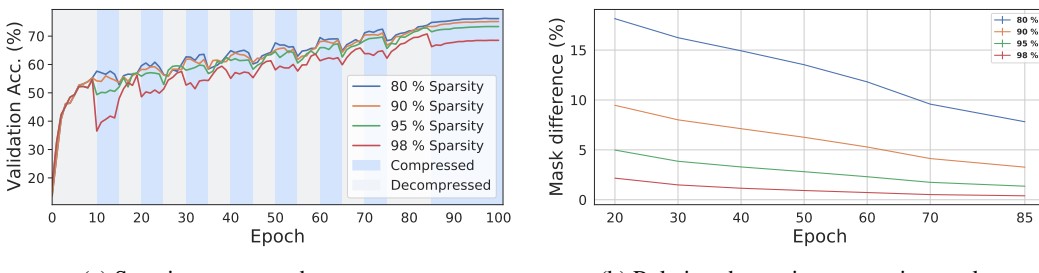

(a) Sparsity pattern and test accuracy

(b) Relative change in consecutive masks

Figure 5: Validation accuracy and sparsity during training, together with differences in consecutive masks for ResNet50 on ImageNet using AC/DC.

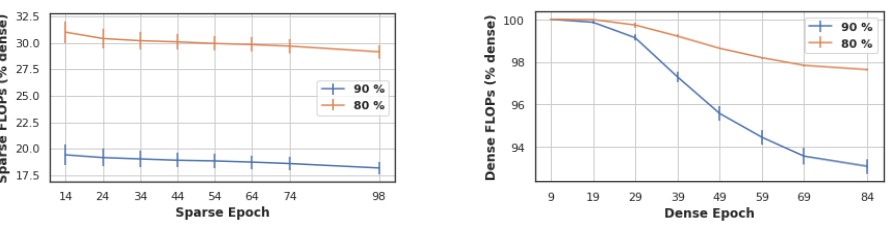

(a) Test FLOPs after each sparse phase

(b) Test FLOPs after each dense phase

Figure 6: Dynamics of sparse and dense inference FLOPs for ImageNet on ResNet50, as a percentage of the dense baseline FLOPs

Moreover, an interesting property of AC/DC is that the resulting dense networks have a small percentage of zero-valued weights, as shown in Table 8. This is most likely caused by "dead" neurons or convolutional filters resulted after each compression phase; the corresponding weights do not get re-activated during the dense stages, as they can no longer receive gradients. This can be easily seen particularly for high sparsity (95% and 98%) where a non-trivial percentage of the weights remain inactive.

**Dynamics of masks and FLOPs during training.** The mask dynamics, measured by the relative change between two consecutive compression masks, have an important influence on the AC/DC training process. Namely, more changes between consecutive compression masks typically imply more exploration of the weights' space, and faster recovery from sub-optimal pruning decision, which in turn results in more accurate sparse models. As can be seen in Figure 5b, the relative mask difference between consecutive compression phases decreases during training, but it is critical to be maintained at a non-trivial level. For completeness, we also included the evolution of the validation accuracy during AC/DC training, for all sparsity levels (please see Figure 5a); at 98% sparsity in particular, it is easiest to see that dense phases enable the exploration of better pruning masks, which ensure that the sparse model improves continuously during training.

Despite the dynamics of the compression masks, we noticed that the sparsity distribution does not change significantly. This can be observed from the number of inference FLOPs per sample, at the end of each compression phase, in Figure 6a. Interestingly, as training progresses, AC/DC also induces structured sparsity, as more neurons and convolutional filters get pruned. This was previously discussed in more detail (see Table 8), but can also be deduced from the decreasing inference FLOPs at the end of each dense phase, as shown in Figure 6b.

Table 8: Accuracy, sparsity, inference FLOPs and percentage of inactive weights for the resulting AC/DC dense models on ResNet50 (before fine-tuning, one seed).

| Target Sparsity | Top-1 Accuracy (%) | Inference FLOPs | Inactive Weights (%) |
|---|---|---|---|
| 80 | 73.8 | 0.98× | 3.2 |
| 90 | 73.1 | 0.93× | 10.5 |
| 95 | 72.9 | 0.85× | 22.0 |
| 98 | 70.8 | 0.67× | 49.8 |

Table 9: AC/DC with uniform vs global magnitude pruning on ResNet50 (one seed), where (⋆) denotes that the first and last layers are dense.

| Sparsity Distribution | Target Sparsity(%) | Global Sparsity(%) | Top-1 Accuracy (%) | FLOPs Inference |
|---|---|---|---|---|
| global | 90 | 89.8 | 75.14 | 0.18× |
| global⋆ | 90 | 82.6 | 75.64 | 0.21× |
| uniform⋆ | 90 | 82.6 | 75.04 | 0.13× |
| global | 95 | 94.8 | 73.15 | 0.11× |
| global⋆ | 95 | 87.2 | 74.16 | 0.13× |
| uniform⋆ | 95 | 87.2 | 73.28 | 0.08× |

**AC/DC with uniform pruning.** As discussed, for example, in [52], global magnitude pruning usually performs better than its uniform counterpart. Interestingly, with global magnitude pruning later layers (which also tend to be the largest) are pruned the most. Moreover, we did not encounter convergence issues caused by entire layers being pruned, as hypothesized in some previous work [16, 32]. However, one concern related to global magnitude pruning is a potential FLOP inefficiency of the resulting models; in theory, this would be a consequence of the earlier layers being pruned the least. For this reason, we performed additional experiments with AC/DC at uniform sparsity, with the first and last layers dense (as commonly used in the literature [16, 32]). Our results show that there are no significant differences compared to AC/DC with global magnitude pruning. However, keeping the first and last layers dense significantly improves the results with global magnitude pruning. These observations emphasize that AC/DC is an easy-to-use method which works reliably well with different pruning criteria. For complete results, please see Table 9.

**Direct comparison with Top-KAST.** As previously highlighted, Top-KAST is the closest to us, in terms of validation accuracy, out of existing sparse training methods. However, for the results reported, the authors kept the first convolutional and final fully-connected layers dense. To obtain a fair comparison, we used AC/DC on the same sparse distribution, and for 90% sparsity over the pruned layers (82.57% overall network sparsity), our results improved significantly. Namely, the best sparse model reached 75.64% validation accuracy (0.6% increase from the results in Table 1), while the accuracy of the best dense model was 76.85% after fine-tuning. For more details, we also provide in Table 10 the results for Top-KAST when all layers are pruned, as they were provided to us by the authors. Notice that AC/DC surpasses even Top-KAST with dense back-propagation.

It is important to note, however, that because of its flexibility in choosing the gradients density, Top-KAST can theoretically obtain significantly better training speed-ups than AC/DC, the latter being constrained by its dense training phases. This allows Top-KAST to improve the accuracy of the models by increasing the number of training epochs, while still enabling (theoretical) training speed-up. We present in Table 11 another comparison between AC/DC and Top-KAST, when the training time for the latter is increased 2 or 5 times; for all results (which were provided to us by the authors), the first and last layers for Top-KAST are dense. When comparing with AC/DC with all layers pruned, Top-KAST obtains better results at 98% and 95% sparsity, with increased training epochs. However, when using the same sparse distribution as Top-KAST (not pruning the first and last layers), the results for AC/DC at 95% and 98% sparsity are significantly better than Top-KAST with increased steps. For all the results reported on AC/DC the number of training steps was fixed at 100 epochs.

We note that the results obtained with AC/DC can be improved as well with increased number of training epochs. As an example, when using the same sparsity schedule extended over 150 epochs, the best sparse model obtained with AC/DC on 90% sparsity reached 75.99% accuracy, using fewer training FLOPs compared to the original dense baseline trained on 100 epochs (namely 87%). Furthermore, when we fine-tune the dense model by replacing the final 15 epochs compression phase with dense training, we obtain a dense model with 76.95% accuracy, higher than the original dense baseline.

## B.3   MobileNet on ImageNet

**Performance of dense models.** Similar to ResNet50, we observed that dense models obtained with AC/DC are able to recover the baseline accuracy after additional fine-tuning. We performed fine-tuning identically to the ResNet50 experiments and observe that AC/DC models obtained with a 75% target sparsity recovered the baseline accuracy, while for 90% the gap is just below 1%. We present results for the (fine-tuned) dense models in Table 7, where (⋆) indicates that the first layer and depth-wise convolutions were never pruned.

**Masks dynamics.** Similar to ResNet50, the change between consecutive AC/DC compression masks plays an important role in obtaining accurate sparse models on MobileNet. As shown in Figure 7b, the compression masks stabilize as training progresses. For completeness, we also illustrate the evolution of the validation accuracy during AC/DC training on MobileNet, at 75% and 90% sparsity, in Figure 7a.

Table 10: Comparison with Top-KAST when pruning all layers (ResNet50)

| Method | Sparsity (%) | Backward Sparsity (%) | Sparse Top-1 Accuracy (%) |
|---|---|---|---|
| **AC/DC** | 80 | 80 / 0 | **76.3 ± 0.1** |
| Top-KAST | 80 | 0 | 75.64 |
| Top-KAST | 80 | 50 | 74.78 |
| Top-KAST | 80 | 80 | 72.19 |
| **AC/DC** | 90 | 90 / 0 | **75.03 ± 0.1** |
| Top-KAST | 90 | 0 | 74.42 |
| Top-KAST | 90 | 50 | 74.09 |
| Top-KAST | 90 | 80 | 73.07 |

Table 11: Comparison with Top-KAST with increased training steps (ResNet50). ($\star$) indicates that the first and last layers are dense for AC/DC, while this is the case for all Top-KAST results.

| Method | Sparsity (%) | Backward Sparsity (%) | Sparse Top-1 Accuracy (%) | Train FLOPs (%) | Inference FLOPs (%) |
|---|---|---|---|---|---|
| **AC/DC** | 80 | 80 / 0 | **76.3 ± 0.1** | 0.65× | 0.29× |
| Top-KAST$_{1\times}$ | 80 | 0 | 75.59 | 0.48× | 0.23× |
| Top-KAST$_{1\times}$ | 80 | 60 | 74.59 | 0.29× | 0.23× |
| Top-KAST$_{2\times}$ | 80 | 0 | 76.11 | 0.97× | 0.23× |
| Top-KAST$_{2\times}$ | 80 | 60 | 75.29 | 0.58× | 0.23× |
| AC/DC | 90 | 90 / 0 | 75.03 ± 0.1 | 0.58× | 0.18× |
| **AC/DC**$^\star$ | 90 | 90 / 0 | **75.64** | 0.6× | 0.21× |
| AC/DC$^\star$ unif. | 90 | 90/0 | 75.04 | 0.55× | 0.13× |
| Top-KAST$_{1\times}$ | 90 | 0 | 74.65 | 0.42× | 0.13× |
| Top-KAST$_{1\times}$ | 90 | 80 | 73.03 | 0.16× | 0.13× |
| Top-KAST$_{2\times}$ | 90 | 0 | 75.35 | 0.84× | 0.13× |
| Top-KAST$_{2\times}$ | 90 | 80 | 74.16 | 0.32× | 0.13× |
| AC/DC | 95 | 95 / 0 | 73.14 ± 0.2 | 0.53× | 0.11× |
| **AC/DC**$^\star$ | 95 | 95 / 0 | **74.16** | 0.54× | 0.13× |
| AC/DC$^\star$ (unif) | 95 | 95 / 0 | 73.28 | 0.5× | 0.08× |
| Top-KAST$_{1\times}$ | 95 | 0 | 71.83 | 0.39× | 0.08× |
| Top-KAST$_{1\times}$ | 95 | 90 | 70.42 | 0.1× | 0.08× |
| Top-KAST$_{2\times}$ | 95 | 0 | 73.29 | 0.77× | 0.08× |
| Top-KAST$_{2\times}$ | 95 | 90 | 72.42 | 0.19× | 0.08× |
| **Top-KAST**$_{5\times}$ | 95 | 0 | **74.27** | 1.94× | 0.08× |
| Top-KAST$_{5\times}$ | 95 | 90 | 73.17 | 0.48× | 0.08× |
| AC/DC | 98 | 98 / 0 | 68.44 ± 0.09 | 0.46× | 0.06× |
| **AC/DC**$^\star$ | 98 | 98 / 0 | **71.27** | 0.47× | 0.08× |
| Top-KAST$_{1\times}$ | 98 | 90 | 67.06 | 0.08× | 0.05× |
| Top-KAST$_{1\times}$ | 98 | 95 | 66.46 | 0.06× | 0.05× |
| Top-KAST$_{2\times}$ | 98 | 90 | 68.99 | 0.15× | 0.05× |
| Top-KAST$_{2\times}$ | 98 | 85 | 68.87 | 0.12× | 0.05× |

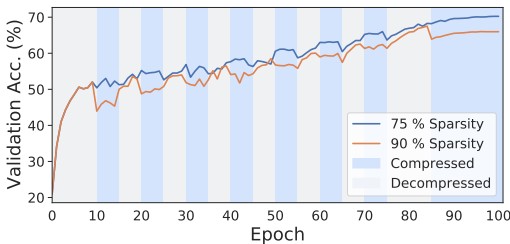 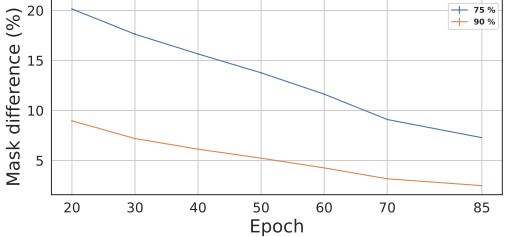

| (a) Sparsity pattern and test accuracy | (b) Relative change in consecutive masks |

Figure 7: Validation accuracy and sparsity during training, together with differences in consecutive masks for ImageNet with MobileNetV1 using AC/DC.

Table 12: Comparison between AC/DC and RigL on MobileNet, where ($\star$) denotes that the first and depth-wise convolutions were kept dense.

| Method | Sparsity (%) | Top-1 Accuracy (%) | Inference FLOPs | Train FLOPs |
|---|---|---|---|---|
| AC/DC | 75 | 70.3 | $0.34\times$ | $0.64\times$ |
| AC/DC$^\star$ | 75 | 70.41 | $0.36\times$ | $0.66\times$ |
| RigL$^\star$ (ERK) | 75 | 68.39 | $0.52\times$ | $0.52\times$ |
| RigL$^\star_{2\times}$(ERK) | 75 | 70.49 | $0.52\times$ | $1.05\times$ |
| RigL$^\star_{5\times}$(ERK) | 75 | 71.9 | $0.52\times$ | $2.63\times$ |
| AC/DC | 90 | 66.08 | $0.18\times$ | $0.56\times$ |
| AC/DC$^\star$ | 90 | 66.56 | $0.21\times$ | $0.58\times$ |
| RigL$^\star$(ERK) | 90 | 63.58 | $0.27\times$ | $0.29\times$ |
| RigL$^\star_{2\times}$(ERK) | 90 | 65.92 | $0.27\times$ | $0.59\times$ |
| RigL$^\star_{5\times}$(ERK) | 90 | 68.1 | $0.27\times$ | $1.47\times$ |

**Comparison with RigL.** We note that the results obtained by RigL [16] improve significantly when increasing the number of training steps 2 or 5 times. Moreover, for all results reported with RigL on MobileNet the first convolutional layer and all depth-wise convolutions are dense, whereas we do not impose such restrictions on our sparse model. Our results can further be improved by using the same sparsity distribution; namely, for 90% sparsity over the pruned parameters (88.57% overall sparsity), the best sparse model obtained with AC/DC achieved 66.56% accuracy (0.5% improvement), while the best dense improved from 67.64% to 70.97% after fine-tuning. In Table 12 we present results for AC/DC and RigL at 75% and 90% sparsity, when the latter is trained over the same number of epochs, or with 2x or 5x the number of passes through the training data. We conclude that AC/DC has very similar validation accuracy to RigL$_{2\times}$. For 75% sparsity, AC/DC achieves similar performance with significantly fewer training and inference FLOPs than RigL. At 90% sparsity, AC/DC and RigL$_{2\times}$ are close in terms of both validation accuracy and training FLOPs; however, the validation accuracy of AC/DC can be improved by almost 0.5% when the first and depth-wise convolutional layers are kept dense. We note that RigL$_{5\times}$ has significantly higher validation accuracy, and for 75% sparsity it even matches the baseline; however, this variant of RigL also uses $2.6\times$ and $1.5\times$ the dense baseline training FLOPs for 75% and 90% sparsities, respectively, which makes it impractical due to its high computational training cost.

## B.4 Inference Speedups

We now examine the potential for real-world speedup of models produced through our framework. For this, we use the CPU-based inference framework of [12], which supports efficient inference over unstructured sparse models, and is free to use for non-commercial purposes. Specifically, we export our Pytorch-trained models to the ONNX intermediate format, preserving weight sparsity, and then execute inference on a subset of samples, at various batch sizes, measuring time per batch. We execute on an Intel i9-7980XE CPU with 16 cores and 2.60GHz core frequency. We simulate two scenarios: the first is *real-time inference*, i.e. samples are processed one at a time, in a resource-constrained environment, using only 4 cores. The second is *batch inference*, for which we pick batch size 64, in a cloud environment, for which we use all 16 cores. We measure average time per batch for the sparse models against dense baselines, for which we use both the Deepsparse engine, and the ONNX runtime (ONNXRT). We present the average over 10 runs. The variance is extremely low, so we omit it for readability.

Table 13: Time per batch (milliseconds) using a sparse inference engine [12].

| Model/Setup | Real-Time Inference, 4 cores | Batch 64 Inference, 16 cores |
|---|---|---|
| ResNet50 ONNXRT v1.6 | 14.773 | 329.734 |
| ResNet50 Dense | 15.081 | 285.958 |
| ResNet50 90% Pruned | 9.46 | 124.193 |
| ResNet50 90% Unif. Pruned | 8.495 | 116.897 |
| MobileNetV1 ONNXRT v1.6 | 2.552 | 80.748 |
| MobileNetV1 Dense | 2.513 | 55.845 |
| MobileNetV1 Pruned 75% | 1.96 | 40.976 |
| MobileNetV1 Pruned 90% | 1.468 | 34.909 |

Table 14: Sample agreement between ResNet50 sparse and dense models

| Method | Sparsity | Sparse Top-1 Accuracy (%) | Dense Top-1 Accuracy (%) | Sparse-Dense Agreement (%) | Sparse-Dense Cross-entropy |
|---|---|---|---|---|---|
| AC/DC | 80% | $76.3 \pm 0.1$ | $76.8 \pm 0.07$ | $89.8 \pm 0.3$ | $0.85 \pm 0.005$ |
| SparseVD | 80% | 75.3 | 75.2 | 98.6 | - |
| GMP | 80% | 76.4 | 76.9 | 86.0 | 1.03 |
| AC/DC | 90% | $75.0 \pm 0.1$ | $76.6 \pm 0.09$ | $86.8 \pm 1.5$ | $1.02 \pm 0.004$ |
| SparseVD | 90% | 73.8 | 73.6 | 98.3 | - |
| GMP | 90% | 74.7 | 76.9 | 83.5 | 1.29 |

We now briefly discuss the results. First, notice that the dense baselines offer similar performance for real-time inference, but that the Deepsparse engine has a slight edge at batch 64. We will therefore compare against its timings below. The results show a speedup of 1.6x for the 90% global-pruned ResNet50 model, and 1.8x for the uniformly pruned one: the uniformly-pruned model is slightly faster, which correlates with its lower FLOP count due to the uniform pruning pattern. This pattern is preserved in MobileNetV1 experiments, although the speedups are relatively lower, since the architecture is more compact. We note that the speedups are more significant for batched inference, where the engine has more potential for parallelization, and our setup uses more cores.

## B.5   Sparse-Dense Output Comparison

To the best of our knowledge, our results are the first ones to show that *both* a dense and a sparse model can be trained jointly. Although other sparse training techniques such as RigL or Top-KAST can train sparse models faster, none of them offer the additional benefit of an accurate dense model.

One method that does generate sparse-dense model couples is Sparse Variational Dropout (SparseVD) [46]; there, after training a dense model with variational inference, a large proportion of the weights can be pruned in a single step, without affecting the accuracy of the dense model. (We note however that Sparse Variational Dropout doubles the FLOP cost of training, due to the variational parameters.) However, our investigation of the SparseVD models trained by [21] shows that the sparse and dense models agree in over 98% of their predictions as measured on the ImageNet validation set, and are of no better quality than the sparse model - if anything, they are slightly worse. Please see Table 14 for complete results.

In comparison, AC/DC with finetuning produces dense models of validation accuracy that is comparable to that of a dense model trained without any compression, and that therefore do differ from their sparse co-trained counterparts. To understand the relative sizes of these differences, we used GMP pruning as the baseline. In particular, we compared the similarity of a fully trained dense model with GMP trained over 100 epochs. We note AC/DC and GMP show comparable accuracy for both their sparse and dense models in this scenario; however, the total training epochs are substantially lower for producing these models with AC/DC.

We use two metrics to investigate the difference in sparse-dense model pairs: the proportion of validation examples on which the top prediction agreed between the two models, and the average cross-entropy of the predictions across all validation examples. In both metrics, Table 14 shows that model similarity is higher for 80% sparsity than 90%, and higher for AC/DC than GMP training: in particular, sparse/dense cross-entropy is about 20% lower for AC/DC, and the number of top-prediction disagreements is about 25% lower for AC/DC at 90% sparsity, and 37% lower for AC/DC at 80% sparsity.

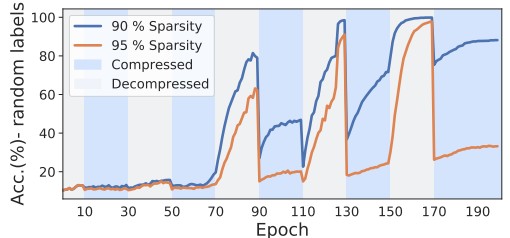 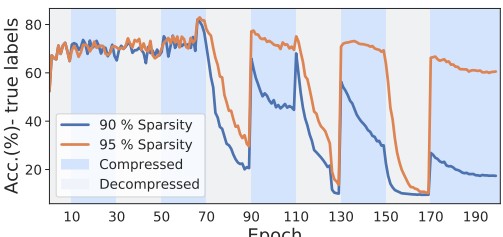

| (a) Accuracy on the mis-labelled data | (b) Accuracy on the mis-labelled data (w.r.t. the true labels) |

Figure 8: Accuracy during training with AC/DC at 90% and 95% target sparsity, for 1000 randomly labelled CIFAR10 images. No data augmentation was applied to the training samples.

### B.6 Memorization Experiments on CIFAR10

In what follows, we study the similarities between the sparse and dense models learned with AC/DC, on the particular setup of memorizing random labels. Specifically, we select 1000 i.i.d. training samples from the CIFAR10 dataset and randomly change their labels. We train a ResNet20 model using AC/DC, at various target sparsity levels, ranging from 50% to 95%. We use SGD with momentum, weight decay, and initial learning rate 0.1 which is decayed by a factor of 10 every 60 epochs, starting with epoch 65.

Using data augmentation dramatically affects the memorization of randomly-labelled training samples, and thus we differentiate between the two possible cases. Namely, the regular baseline can easily memorize (in the sense of reaching perfect accuracy) the randomly-labelled samples, when *no* data augmentation is used; in comparison, with data augmentation memorization is more difficult, and the accuracy on randomly-labelled samples for the baseline is just above 60%. In addition to the accuracy on the perturbed samples with respect to their new random labels, we also track the accuracy with respect to the "true" or correct labels. This differentiation offers a better understanding regarding where memorization fails and a glimpse into the robustness properties of neural networks in general, and of AC/DC, in particular.

**No data augmentation.** As previously mentioned, in this case the baseline model can perfectly memorize the perturbed data, with respect to their random labels. Interestingly, prior to the initial learning rate decay, most ($\geq 70\%$) perturbed samples are still correctly classified with respect to their "true" labels, and memorization happens very quickly after the learning rate is decreased. In the case of AC/DC with low target sparsity (50% and 75%), memorization has a very similar behavior to the dense baseline. However, for higher sparsity levels (90% and 95%) we can see a clear difference between the sparse and dense models. Namely, during each compression phase most perturbed samples are correctly classified with respect to their true labels, whereas in decompression phases their random labels are memorized. This phenomenon is illustrated in Figure 8.

**Data augmentation.** In this case, memorization of the perturbed samples is more difficult, and it happens later on during training, usually after the second learning rate decrease for the baseline model. Interestingly, in the case of AC/DC we can see (Figure 9) a clear inverse relationship between the amount of memorization and the target sparsity. Although low sparsity enables more memorization, most perturbed samples are still correctly classified with respect to their true labels. For higher sparsity levels (90% and 95%), most perturbed samples are correctly classified with respect to their true labels (almost 90%) and very few are memorized. Furthermore, the dense model resulted from AC/DC training is more robust than the original baseline, as it still learns the correct labels of the perturbed samples, despite being presented with random ones.

## C   Computational Details

### C.1   Hardware Details

Experiments were run on NVIDIA RTX 2080 GPUs for image classification tasks, and NVIDIA RTX 3090 GPUs for language modelling. Each ImageNet run took approximately 2 days for ResNet50 and one day for MobileNet, while each Transformer-XL experiment took approximately 2 days.

### C.2   FLOPs Computation

When computing FLOPs, we take into account the number of zero-valued weights for linear and convolutional layers. To compute the FLOPs required for a backward pass over a sample, we use the same convention as RigL [16]; namely, if $F$ denotes the inference FLOPs per sample, the number of backward FLOPs is estimated as $B = 2 \cdot F$, as we need $F$ FLOPs to backpropagate the error, and additional $F$ to compute the gradients w.r.t. the

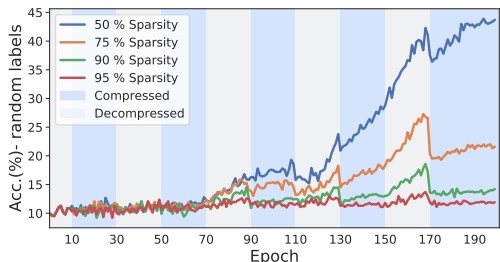

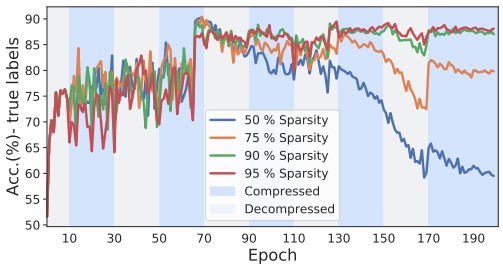

(a) Accuracy on the mis-labelled data

(b) Accuracy on the mis-labelled data (w.r.t. the true labels)

Figure 9: Accuracy during training with AC/DC at 50%, 75%, 90% and 95% target sparsity, for 1000 randomly labelled CIFAR10 images. Here, all samples were trained using data augmentation.

weights. For ImageNet experiments, we ignore the FLOPs required for Batch Normalization, pooling, ReLU or Cross Entropy, similarly to other methods [16, 52, 37]; however, these layers have a negligible impact on the total FLOPs number (at most $0.01\times$ the dense number).

For compression and decompression phases $C$ and $D$, we consider $F_C$ and $F_D$ the compression and decompression inference FLOPs per sample, respectively. We use $F$ to denote the inference FLOPs per sample for the baseline network. During each compression phase, the training FLOPs per sample can be estimated as $3 \cdot F_C$. For decompression phases, we noticed that a small fraction of weights remain zero, and therefore $F_D < F$. When doing a backward pass we have additional $F_D$ from back-propagating the error, and $F$ extra FLOPs for the gradients with respect to all parameters. Therefore, we estimate the training FLOPs per sample during a decompression phase as $2 \cdot F_D + F$. We measure the number of FLOPs on a random input sample, at the end of each training epoch and use this value to estimate the total training FLOPs for that particular epoch. To obtain the final number of FLOPs, we compute the inference FLOPs on a random input sample, estimate the backward FLOPs, compute the estimated training FLOPs over all training epochs as described above, and scale by the number of training samples.

## C.3    Choice of Hyper-parameters

**Length of compression/decompression phases.** AC/DC alternates between compression and decompression phases to co-train sparse and dense models. It is important to note, however, that the length of these phases, together with the warm-up and fine-tuning phases, could have a significant impact on the quality of the resulting models. Before settling on the sparsity pattern we used for all our ImageNet experiments (see Figure 5a and Figure 7a), we experimented with different lengths for the sparse/dense phases, but found that ultimately the pattern used in the paper had the best trade-off between training FLOPs and validation accuracy.

Notably, we experimented on ResNet50, 90% sparsity, with increasing the training epochs to 130 and with different lengths for the compression/decompression phases. For example, we found that alternating between sparse/dense training every 10 epochs yielded slightly better results after 130 epochs: $75.34\%$ for the sparse model, $76.87\%$ for the fine-tuned dense model; however, this also had higher training FLOPs requirements ($0.7\times$ for the sparse model and $0.9\times$ including the fine-tuned dense). We additionally experimented with longer dense phases (10 epochs), compared to sparse phases (5 epochs); this also resulted in more accurate models: $75.45\%$ accuracy for the sparse model and $76.78\%$ – for the fine-tuned dense model. However, the training FLOPs were substantially higher: $0.85\times$ for the sparse model and $1.15\times$ for the fine-tuned dense.

Due to computational limitations, and to ensure a fair comparison with the dense baseline and other pruning methods, we decided on using a fixed number of 100 training epochs (the same used for the dense baseline). In this setup, we experimented mainly with the lengths for the compression/decompression phases used in Figure 5a and Figure 7a, but noticed that having a longer final decompression phase had a positive impact on the fine-tuned dense model. For instance, when following a sparsity schedule as in Figure 1, the sparse model at 90% sparsity had a very similar performance to the reported results (75.18% accuracy, from one seed), while the fine-tuned dense model was significantly below the dense baseline (76.05% validation accuracy). We believe having a short warm-up period and a longer fine-tuning phase are both beneficial for the sparse model; in our experiments, we only used warm-up phases of 10 epochs, but believe that shorter phases are worth exploring as well. Furthermore, the mask difference between consecutive compression phases is an important guide for choosing the sparsity schedule: as it was previously discussed, having a non-trivial difference between the masks typically results in better sparse models. Illustrations of the pruning masks during training on ImageNet are presented in Figure 5b and Figure 7b.

When choosing the sparsity schedule for the language models experiments on Transformers-XL, we followed the same principles as for ImageNet. In fact, the sparsity schedule is very similar to the one used for ImageNet, scaled by the number of training epochs ( 48 epochs or 100,000 steps for Transformers).

In the case of CIFAR100, we used for AC/DC the same number of 200 training epochs as for the dense baseline. We experimented with sparse/dense phases of lengths 10 or 20, and found that generally switching every 20 epochs between sparse and dense training yielded the best results.

**Training Hyper-parameters for ImageNet.** We used the same hyper-parameters for all our ImageNet experiments, on both ResNet50 and MobileNetV1. Namely, we trained using SGD with momentum and batch size 256. We used a cosine learning rate scheduler, after an initial warm-up phase of 5 epochs, when the learning rate was linearly increased to 0.256. The momentum value was 0.875 and weight decay was 0.00003051757813. These hyper-parameters have the standard values used in the implementation of STR [37]. Furthermore, to improve efficiency, we train and evaluate the models using mixed precision (FP16). For models trained with mixed precision, the difference in accuracy between evaluating them with FP32 versus FP16 is negligible (<0.05%). However, we noticed larger differences (around 0.2-0.3%) in accuracy when training AC/DC with FP16 versus FP32.

**Training Hyper-parameters for Transformer-XL.** For our Transformer-XL experiments, we integrated into our code-base the implementation provided by NVIDIA [2], which also follows closely the original implementation in [10]. We used the same hyper-parameters for training the large Transformer-XL model with 18 layers on WikiText-103, including the Lamb optimizer [57] with cosine learning rate scheduler. For these experiments, we did not reset the statistics involving exponential moving average of gradients before entering a dense phase.

---

[2]`https://github.com/NVIDIA/DeepLearningExamples/tree/master/PyTorch/LanguageModeling/Transformer-XL`