# OpenReview forum: "AC/DC: Alternating Compressed/DeCompressed Training of Deep Neural Networks"
_NeurIPS.cc/2021/Conference — NeurIPS 2021 Poster_

### Official Review · Reviewer_74zE · 2021-07-11

**Rating:** 6
**Confidence:** 3

**Summary:**

The paper proposes a sparse training technique for DNNs. Unlike previous post-training or sparse-only training, AC/DC algorithm alters the phase of dense and sparse phase. There have been several altering approaches (ex: DSD), but the paper aims to provide both sparse and dense models. The algorithm targets unstructured (or, semi-structured) sparsity. Pruned elements are selected by the global magnitude threshold at the start of the sparse phase. Experiments on image classification and language modeling show that AC/DC training can achieve comparable or better results compared to previous works (RigL, Top-KAST) on sparse models and the dense baseline.

**Limitations And Societal Impact:**

The authors well addressed the limitations of the proposed method. (… difficult practical implementation, still dense phase exists, …) Because the idea mainly focuses on efficient training, I do not think this paper needs more discussion about the social impact.

**Main Review:**

The idea is simple and looks promising. The motivation and algorithm are clear and easy to follow.

Comments:

(1)	The idea of altering sparse/dense is not new, but the practical issues (ex: gradient momentum, altering period, …) are handled well, so I think this paper achieved some level of novelty in terms of implementation.

(2)	It seems that after dense fine-tuning, the result almost recovers the dense baseline (line 310-312). It will help readers more if the dense comparison is also provided in tables.

(3)	I am curious that why the dense phase is ‘necessary’. For the selection of sparse/dense phase length (Appendix C.2), I think we can change the number of epochs while maintaining FLOPs. For example, longer the sparse phase and reduce the dense phase while keeping the FLOPs to see how much dense updates are required at least.

Weaknesses/Questions:

(1)	I think there should be some explanations on ‘why’ the AC/DC benefits more in higher sparsity than medium sparsity, compared to post-training techniques. In addition, ‘why’ fine-tuned dense sometimes fail to recover (table 4).

(2)	How is the ‘sample-level agreement’ related to the effectiveness of AC/DC? Also, there seems to be not much difference (90% vs 86%, line 378-379).

I appreciate that the authors explained the related works and experiment settings in detail. Please note that, because I am not familiar with theoretical parts, I cannot give an opinion as to whether the theoretical analysis is correct.

--------
After reading the authors' response, I think my concerns are well addressed.
I appreciate authors, especially for extra experiments, and I'll keep my score unchanged.

**Time Spent Reviewing:**

5

---

> ### Author Response · Authors · 2021-08-10
> **Response to Reviewer 74zE - Reducing FLOPs Count**
>
> Thank you for the constructive review! Please find below our answers, with reference to each specific question or comment.
>
> ---
>
> >It seems that after dense fine-tuning, the result almost recovers the dense baseline (line 310-312). It will help readers more if the dense comparison is also provided in tables.
>
> Indeed, with our method we can almost recover the dense baseline, with additional fine-tuning. The results for the dense models (before and after fine-tuning) trained on ImageNet, can be found in the Appendix, in Table 6 for ResNet50, and Table 7 for MobileNetV1. We will provide a more explicit reference to these results in the main body of the paper.
>
> ---
>
>
> > I am curious that why the dense phase is ‘necessary’. For the selection of sparse/dense phase length (Appendix C.2), I think we can change the number of epochs while maintaining FLOPs. For example, longer the sparse phase and reduce the dense phase while keeping the FLOPs to see how much dense updates are required at least.
>
> This is a very interesting point. Our theoretical analysis seems to suggest that dense phases are necessary for obtaining good convergence guarantees. Furthermore, our empirical analysis shows that dense phases are crucial to obtaining accurate dense models at the end of training. Since the pruning masks for the subsequent sparse phases are computed based on the weights of the dense model, if the latter is too weak, then this might result in sub-optimal pruning decisions, which in turn will lead to less accurate sparse models.
>
> We followed your suggestion of decreasing the length of the dense phases and performed experiments on both CIFAR100 and ImageNet. For CIFAR100, we decreased the length of dense phases, while keeping the FLOPs count constant (by increasing the number of training epochs); our results show that for intermediate sparsity (50% and 75%) the accuracy increased slightly, while for higher sparsity levels, results were marginally worse than the previous ones. Please find the plot with these results here: https://github.com/neurips-acdc-5828/acdc-plots/blob/main/CIFAR100_pruning_performance.png . In the case of ImageNet and ResNet50, we obtained 25-30% reduction in training FLOPs, by decreasing the dense phases length to less than half, at the expense of 0.3-0.4 reduction in accuracy, for both 90% and 95% sparsity.
>
> ---
>
> > I think there should be some explanations on ‘why’ the AC/DC benefits more in higher sparsity than medium sparsity, compared to post-training techniques. In addition, ‘why’ fine-tuned dense sometimes fail to recover (table 4).
>
> Our intuition on this point is that, given a model density budget, AC/DC trains a set of “robust sparse features” according to that budget, from scratch. This can be seen for example in Table 8 in the Appendix, which shows that the dense models obtained at the end of training have a relatively small percentage of the weights at zero, due to the pruning of entire filters or neurons during sparse training. This is different from post-training techniques, where the sparse features are inherently derived from pruning the dense features. The fine-tuning phase provides in some sense the “inverse” process, by which sparse features are augmented by re-introduction to better fit the data, especially hard examples.
>
> We conjecture that the issue at high sparsity is that sometimes the sparse features are too lossy. However, this may also be fixed by additional fine-tuning. For example, at 95% sparsity, extending the fine-tuning time of the dense model to 30 epochs almost recovered the baseline accuracy; namely, we obtained 76.48% accuracy, slightly below the 76.84% baseline. Regarding Table 4, we would like to acknowledge that Transformers are prone to overfitting, which might have a negative influence on fine-tuning. Furthermore, we also believe that exploring more options for the sparse and dense phases lengths could improve the perplexity scores. We will look more into this issue in future work.
>
> ---
>
> > How is the ‘sample-level agreement’ related to the effectiveness of AC/DC? Also, there seems to be not much difference (90% vs 86%, line 378-379).
>
> The ‘sample-level agreement’ is not related to the effectiveness of our method, but we do believe it sheds some light on how the sparse and dense models are related, in the context of co-training. Although the differences appear small when viewed this way, clearly the baseline would also have to have high overlap in order to be accurate; a better view would be in terms of the remaining error (10% vs 14%), which suggests that one would need to disagree on almost 50% more samples than the other.

---

### Official Review · Reviewer_4fim · 2021-07-12

**Rating:** 7
**Confidence:** 4

**Summary:**

The authors propose AC/DC, a method that performs co-training of sparse and dense neural networks. The proposed method builds on iterative hard thresholding (IHT) methods and the authors include theoretical analysis of the convergence properties of their algorithm. The authors perform an extensive empirical analysis of their technique against existing methods.


**Ethical Concerns:**

There are no ethical issues with this paper.

**Limitations And Societal Impact:**

I did not identify potential negative societal impacts in this work.

**Main Review:**

Strengths

The paper is well written and does a good job of introducing related work and background necessary to understand the work.
The authors work to include theoretical analyses along with empirical results. In addition to practical efficiency studies, the authors study interesting properties of their models (i.e., memorization in sparse and dense models)

Weaknesses

The experiments are extensive in some sense but it’s very difficult to draw an apples to apples comparison between this method and others with the provided data and presentation. I want to highlight that much of this difficulty is not unique to this paper; It’s an issue across all sparsity literature that plagues any paper on the subject. My guidance here will not perfectly solve this problem, but I hope that it will help to present a clearer picture.

First, the conclusions that can be drawn vary across training and inference. For example, in Table 1, AC/DC gets 1.2% higher top-1 (@ 80% sparsity) at the cost of 50% more training computation but with 30% lower inference computation. At these sparsities AC/DC had lower top-1 than WoodFisher 35% higher inference cost. None of the pairs of accuracy, inference flops or training flops match, so it’s hard to reason through the comparison. I’d much prefer to see flop-accuracy efficiency curves plotted for all methods for both training and inference. This would allow for a much cleaner comparison. It would also allow you to incorporate a broader set of results from other methods, e.g. longer training run results with RigL/Top-KAST, different backward sparsities for Top-KAST. I’d recommend plotting all results from RigL/Top-KAST in this way for all variants and drawing a line for the most efficient frontier they achieve across all variants.

Some smaller notes to improve the experiments:
- Please include WoodFisher inference costs in Table 2.
- AC/DC does not exactly match sparsity. This becomes less negligible at 98% sparsity, where your AC/DC models have 10% (2.2% v. 2% density) more parameters than the baselines.
- Speedup claims over dense baselines should be for the same accuracy. Claiming speedups with lower accuracy sparse models is not a useful or fair comparison.
- It’d be nice to see the inclusion of some during-training sparsification results, e.g. the magnitude pruning results from “The State of Sparsity”. Technically speaking, these methods could also achieve training speedups by exploiting sparsity after the % of dead weights becomes sufficiently high.
- For RigL, non-ERK results would be good to include because I believe they’re more efficient in terms of accuracy per flop.
- For Top-KAST, please specify what backward sparsity was used for each result listed in the tables.

Comments
- I am not qualified to validate the theorems and proofs in this work.
- In the “Related Work” section: post-training and sparse training are not the only techniques for sparsification. During-training sparsification is worth consideration, e.g. the pruning method introduced in “To Prune or Not to Prune”.
- The authors should call out the ImageNet dataset in Table 1, 2, and 3. I had to read through more of the text past where the tables are to discover which dataset these results were for.
- Claim about “little to no training speedup” in Top-KAST with 0% backward sparsity is an exaggeration. 2/3rds of the linear operations would still be sparse. It’s better to compare with quantitative results (i.e., include FLOPs in Table 4).

Summary

Overall this is a good paper that is well done and is a valuable contribution to the field. I believe it should be accepted, but I also feel strongly that the authors should work to address some of the limitations in their experiments section to provide the most accurate and useful comparison for the reader.

[UPDATE POST-REBUTTAL]

I think the paper will be very solid once the author's clarifications and additions are made. This paper makes a valuable contribution to the field and I think it should be accepted.

**Time Spent Reviewing:**

2hrs

---

> ### Author Response · Authors · 2021-08-10
> **Response to Reviewer 4fim - Comparisons with Previous Work on Sparse Training**
>
> Thank you for the comments and suggestions! We addressed each question or comment below.
>
> ---
>
> > I’d recommend plotting all results from RigL/Top-KAST in this way for all variants and drawing a line for the most efficient frontier they achieve across all variants.
>
> Thank you for the suggestion! We acknowledge the difficulty in comparing across multiple methods. Here is the suggested plot for uniform pruning at 90% sparsity, where we also included some additional AC/DC runs with shorter dense phases: https://github.com/neurips-acdc-5828/acdc-plots/blob/main/acc_flops_90sparsity.png
>
> ---
>
> > Please include WoodFisher inference costs in Table 2.
>
> We will do so in the next revision of the paper.
>
> ---
>
> >AC/DC does not exactly match sparsity. This becomes less negligible at 98% sparsity, where your AC/DC models have 10% (2.2% v. 2% density) more parameters than the baselines.
>
>
> The sparsity levels presented in Tables 1, 2 and 3 are computed with respect to all the parameters of the network, including the biases and the Batch Normalization parameters, which are not pruned. This results in a 0.2%  lower real sparsity level compared to the target sparsity level. However, as far as we know, the sparsity levels presented in other papers, such as RigL and Top-KAST, are computed only with respect to the prunable parameters, and therefore the real sparsity levels match ours. We will update the sparsity numbers accordingly in a future revision of the paper, and add the appropriate clarifications.
>
> ---
>
> >Speedup claims over dense baselines should be for the same accuracy. Claiming speedups with lower accuracy sparse models is not a useful or fair comparison.
>
> We agree that this is true for sparse models, which typically do not match the baseline accuracy. However, we believe for the resulting dense models the comparison is fair, since at least for 80% and 90% sparsity we recover the baseline accuracy, and use fewer training FLOPs (including fine-tuning). For example, as described in lines 325-326, the fine-tuned dense model obtained with 90% target sparsity achieves 76.56% accuracy (slightly lower than the 76.84% baseline), but with 27% fewer training FLOPs. We additionally ran our method for an extended training time. For 1.5 x training, at 90% global sparsity, our best sparse model achieves 75.99% accuracy, with 87% of the training FLOPs of the dense baseline. Furthermore, at 2x training and using uniform pruning, our best sparse model achieves 76.1% accuracy (at 90% sparsity), and 74.3% accuracy (at 95% sparsity), but at the expense of a similar number of training FLOPs to the dense baseline.
>
> ---
>
> > It’d be nice to see the inclusion of some during-training sparsification results, e.g. the magnitude pruning results from “The State of Sparsity”. Technically speaking, these methods could also achieve training speedups by exploiting sparsity after the % of dead weights becomes sufficiently high.
>
> We will work these results into the next revision. Considering the results in Figure 2 from the RigL paper, gradual magnitude pruning can indeed achieve some training speed-ups, with a during-training sparsification strategy. However, we note that for a similar training budget, we obtain more accurate sparse models with AC/DC. For example, on ResNet50 at 90% sparsity, magnitude pruning with a training budget of 0.51x baseline FLOPs achieves 73.9% accuracy, whereas AC/DC for a similar budget (0.55 x baseline FLOPs) has a considerably higher accuracy (75.04%). Moreover, we emphasize that AC/DC has the advantage over other pruning methods of also co-training accurate dense models.
>
> ---
>
> > For RigL, non-ERK results would be good to include because I believe they’re more efficient in terms of accuracy per flop.
>
> Agreed, we will include these results in a separate row.
>
> ---
>
> > For Top-KAST, please specify what backward sparsity was used for each result listed in the tables.
>
> We used 50% backward sparsity for the 80% sparse model, and 80% backward sparsity for 90% forward sparsity (from Table 1 in the Top-KAST paper), while for both 95% and 98% forward sparsity, the results are obtained using 90% backward sparsity.
>
> ---
>
> > In the “Related Work” section: post-training and sparse training are not the only techniques for sparsification. During-training sparsification is worth consideration, e.g. the pruning method introduced in “To Prune or Not to Prune”.
>
> Thank you for the reference, we will discuss it in a future version of the paper. We would like to point out, however, that we discuss several during-training sparsification methods in the paragraph starting from line 141, such as, for example, “Dynamic model pruning with feedback”.
>
> ---
> >The authors should call out the ImageNet dataset in Table 1, 2, and 3. I had to read through more of the text past where the tables are to discover which dataset these results were for.
>
> We agree and we will make the reference to ImageNet more explicit.
>
> ---
>
> >Claim about “little to no training speedup” in Top-KAST with 0% backward sparsity is an exaggeration. 2/3rds of the linear operations would still be sparse. It’s better to compare with quantitative results (i.e., include FLOPs in Table 4).
>
> We agree and we will definitely revise this claim.

---

> > ### Comment · Reviewer_4fim · 2021-08-13
> > **Reviewer Response**
> >
> > Thank you for your reply. I think the paper will be very solid once the above clarifications and additions (in particular, the pareto frontier charts) are made. This paper makes a valuable contribution to the field and I think it should be accepted.

---

### Official Review · Reviewer_4BMG · 2021-07-15

**Rating:** 5
**Confidence:** 3

**Summary:**

The paper proposes a method to reduce the (theoretical) training FLOPs by alternating between dense and sparse training. For specific assumptions, the paper also presents a convergence result that supports the use of the presented method. The method achieves higher accuracy than other sparse-training baselines (albeit using more training FLOPS) and achieves higher accuracy than other post-training pruning methods for sparsity >= 95%.

**Limitations And Societal Impact:**

Sufficiently addressed (limitations: see above)

**Main Review:**

The paper is well written. In my opinion, the practical significance of the presented method is limited since the reported theoretical training FLOPs do not (currently) seem to translate to real training speed-ups (this is a main goal of the paper; see line 163). However, the proposed method seems to outperform state-of-the-art methods relying on post-training pruning at high sparsity levels. I wonder why there are severe differences between inference FLOPSs of different models at the same sparsity levels. For instance, in Table 2 at 98% sparsity AC/DC requires three times as many FLOPs as STR. Can you please comment on that.

The method uses unstructured sparsity across all layers. How does sparsity distribute across layers? For instance, is there some difference between input/output layers and intermediate layers?

If I interpret Theorem 1 correctly, it appears that the method convergences to a point where we can expect the error to be lower than 16 \sigma^2 / \alpha (plus some \epsilon that I ignore for the moment). Assuming that the assumptions hold, can we expect this bound to be meaningfully small in practice (I could think of either \sigma being very large or \alpha being very small), or is the main benefit that we are guaranteed to not be arbitrarily far away from the minimum when using the proposed method? At least it appears that the bound depends heavily on the gradient variance. This implies that averaging several stochastic gradients to reduce the gradient variance (e.g., larger mini-batches) would also result in a better bound. It would be interesting to test this in a synthetic setting.

It is being discussed that the proposed method serves as a regularizer (wrongly labeled samples cannot be learned) but runs the risk of preventing memorization of "hard samples", which may harm accuracy. This suggests that some intermediate sparsity level could result in improved accuracies. It would be interesting to see if there exists (at least on some datasets) an intermediate sparsity level that results in better accuracies than an unconstrained dense model.

What do we learn from the fact that there is higher sample-level agreement for co-trained sparse-dense pairs than for post-training pruning sparse-dense pairs? I am not sure what to make out of this, besides viewing it as an interesting observation.

The paper uses the word "significant" several times. Did you perform any kind of statistical test to verify this? Otherwise, I suggest to refrain from using the word "significant".

**Time Spent Reviewing:**

6

---

> ### Author Response · Authors · 2021-08-10
> **Response to Reviewer 4BMG - Clarifications on the Practical Potential of AC/DC**
>
> We would first like to thank the reviewer for the comments and suggestions. Please see below our answers, with reference to each specific comment or question.
>
> ---
> >The paper is well written. In my opinion, the practical significance of the presented method is limited since the reported theoretical training FLOPs do not (currently) seem to translate to real training speed-ups (this is a main goal of the paper; see line 163). However, the proposed method seems to outperform state-of-the-art methods relying on post-training pruning at high sparsity levels.
>
> Thank you for your comment. We acknowledge this limitation, but please note that this would apply equally well to other work in the area, e.g. RigL and Top-KAST. At the same time, there is emerging hardware and software support for unstructured sparsity in the context of sparse training, with already existing support in the context of inference. One specific instance of such support is the PopSparse library, which works with Graphcore IPUs. In our preliminary exploration using a simple instance of MLPs for MNIST, on a Graphcore IPU Mark 1, we have observed that training in the 95% sparse support (static, unstructured sparsity) resulted in 20% clock time savings, compared to a fully dense model. While currently these savings are quite modest, our findings show the potential of using sparse operations to accelerate model training.
>
> Furthermore, even with the current hardware limitations, we do provide a training “speed-up” compared to post-training pruning methods. Namely, for a similar number of training epochs, we obtain both a dense and a sparse model, a property not found in other pruning methods, to our knowledge.
>
> ---
>  > I wonder why there are severe differences between inference FLOPSs of different models at the same sparsity levels. For instance, in Table 2 at 98% sparsity AC/DC requires three times as many FLOPs as STR. Can you please comment on that. The method uses unstructured sparsity across all layers. How does sparsity distribute across layers? For instance, is there some difference between input/output layers and intermediate layers?
>
> The answers to these two questions are linked. We employ global magnitude across all layers when pruning in the basic version of the algorithm, which can result in uneven sparsity distributions across layers. Specifically, we found that the method prunes the input and output layers much less relative to the average sparsity. Also, we noticed that, in general, earlier layers are pruned at a lower level, while wider convolutional layers tend to have a higher than average sparsity. We provide a plot to illustrate the density distribution across the layers of ResNet50, trained on ImageNet, at 80% and 90% global sparsity: https://github.com/neurips-acdc-5828/acdc-plots/blob/main/ImagenetDensityPlot.pdf . This non-uniform sparsity distribution results in different FLOPs relative to e.g. STR for the sparse model, as STR has a more “uniform” sparsity profile across the layers.
>
> We note however that 1) AC/DC shows good results with arbitrary sparsity profiles, in particular uniform ones, and 2) sparse STR and AC/DC models lead to fairly similar inference times, despite the FLOP difference. (This is because inference performance depends on many other factors, such as layer sizes and activation density.) Please see Table 13 in the Appendix for numbers at 90% sparsity (STR has practically identical inference times to the 90% uniformly pruned model, leading to a 10% difference in terms of the inference time of sparse models).
>
> ---
> > If I interpret Theorem 1 correctly, it appears that the method convergences to a point where we can expect the error to be lower than 16 \sigma^2 / \alpha (plus some \epsilon that I ignore for the moment). Assuming that the assumptions hold, can we expect this bound to be meaningfully small in practice (I could think of either \sigma being very large or \alpha being very small), or is the main benefit that we are guaranteed to not be arbitrarily far away from the minimum when using the proposed method? At least it appears that the bound depends heavily on the gradient variance. This implies that averaging several stochastic gradients to reduce the gradient variance (e.g., larger mini-batches) would also result in a better bound. It would be interesting to test this in a synthetic setting.
>
> The AC/DC algorithm is an example of a practical algorithm whose design is informed by theory. One of our contributions is to show that a version of the IHT method, which is common in the sparse recovery literature, carries over to the more general setting of sparse NN training. For both IHT and AC/DC, the theoretical guarantees are more pessimistic than what is experimentally verified, which is partially due to the worst-case analysis. In fact, our analysis tolerates a more robust dependence in parameters, in the sense that it allows the CPL inequality to be violated a certain number of times during the execution of the algorithm (whenever we reach nearly stationary points), as long as the expected loss still decreases.
>
> Our analysis is completely generic, and as mentioned by the reviewer, we can reduce sigma by taking larger batches (specifying the theoretical guarantees for a generic sigma, without being specifically concerned with batch size and other variance reduction tricks is standard in the optimization literature). While we could increase batch size, it is known that doing so affects generalization. Hence we chose a standard batch size, which exhibits good empirical behavior. Understanding the trade-off between mini-batch size, training error and test error would shed more light on this matter.
>
> ---
>
> > It is being discussed that the proposed method serves as a regularizer (wrongly labeled samples cannot be learned) but runs the risk of preventing memorization of "hard samples", which may harm accuracy. This suggests that some intermediate sparsity level could result in improved accuracies. It would be interesting to see if there exists (at least on some datasets) an intermediate sparsity level that results in better accuracies than an unconstrained dense model.
>
> Thank you for the suggestion! We agree, our method can act as a regularizer and result in higher accuracy than the baseline in case of intermediate sparsity. In fact, we do have experiments to confirm this. For example, at 50% sparsity, the sparse model trained with AC/DC on ImageNet for ResNet50 achieves 77.05% accuracy, higher than the 76.84% dense baseline. Furthermore, our CIFAR100 experiments (please see Table 5 in the Appendix) further confirm this regularizing effect; namely, in this case the accuracy for the sparse models at 50% and 75% sparsity is higher than the dense baseline.
>
> ---
> > What do we learn from the fact that there is higher sample-level agreement for co-trained sparse-dense pairs than for post-training pruning sparse-dense pairs? I am not sure what to make out of this, besides viewing it as an interesting observation.
>
> This behavior is related to co-training; we view it as a potentially useful side-effect of the method, when we are interested in the difference in predictions between the sparse and dense models. For example, in constrained environments where sparse models are needed, it would be useful to know that the predictions would be similar to the ones resulting from their dense counterparts. This could be especially desirable in the context of adversarially-robust training.
>
> ---
> >The paper uses the word "significant" several times. Did you perform any kind of statistical test to verify this? Otherwise, I suggest to refrain from using the word "significant".
>
> Indeed, the word “significant” is not used in the statistical sense here, and we will be more careful about the use of this term. However, we do believe that the accuracy our method obtains is in fact higher (particularly in the high sparsity regimes) than the one reported in previous works, such as RigL or WoodFisher, in the statistical sense. For example, at 95% sparsity on ResNet50, AC/DC achieves 1% higher accuracy than the next best method (WoodFisher), with a standard deviation of 0.2 across the 3 different runs. This difference is even more apparent at 98% sparsity, where the gap between AC/DC and the next best method (Top-KAST) is even higher, and similarly for MobileNetV1 at 90% sparsity.

---

> > ### Comment · Reviewer_4BMG · 2021-08-30
> > **Response**
> >
> > I would like to thank the authors for their response that addressed my questions and concerns. I stress that the issue regarding practical relevance does not negatively affect my rating and I appreciate work on sparse models even though practical implications are limited. I appreciate your comments and clarifications on practical aspects. After reading the rebuttal I keep my rating.

---

### Author Response · Authors · 2021-08-10
**Overview of Authors Response**

We would like to thank the reviewers for their high-quality feedback, and for the very interesting questions! Overall, the reviewers seemed to appreciate the writing and the results, and had many detailed questions in particular with respect to:
1. Practical potential - Reviewer 1 (4BMG)
2. Comparisons with previous work on sparse training - Reviewer 2 (4fim)
3. Reducing FLOP counts - Reviewer 3 (74zE)

We briefly overview our responses:
1. On this point, AC/DC has similar characteristics to prior work on sparse training. Moreover, we present very preliminary results suggesting that sparse training speedups can be obtained on Graphcore IPUs, and point out that the sparse models we obtain do execute with inference speedups (as shown in Appendix Table 13).
2. We have provided several clarifications on these points, including the graphs suggested by the reviewer.
3. Following the reviewer’s suggestion, we have experimented with shortening the length of the dense phases. Surprisingly, we found that this leads to relatively low accuracy loss, but reduces total training flops by a further 25-30%.

We also provide detailed responses to all the reviewers questions. We look forward to engaging with them during the discussion!

---

### Decision · Program_Chairs · 2021-09-27

**Decision:**

Accept (Poster)

**Comment:**


The paper suggests a network compression scheme which alternates, in training, between dense (uncompressed) phases and sparse (compressed) phases.  The compressed phases work along the lines of stochastic IHT.  The paper’s merits are mostly in the experimental side.  There is also a theoretical part which in my opinion does not directly explain the experimental success but is still somewhat relevant to the main idea.  The reviewers seem to have done a thorough review and seem to slightly lean toward acceptance.  Given the importance of the field of network compression together with the merits of this paper, I am inclined to accept.